# GUARANTEED GENERATION FROM LARGE LANGUAGE MODELS

**Minbeom Kim**[1*†]    **Thibaut Thonet**[2]    **Jos Rozen**[2]    **Hwaran Lee**[3,4]    **Kyomin Jung**[1]
**Marc Dymetman**[5†]

[1]Seoul National University  [2]NAVER Labs Europe  [3]NAVER AI Lab  [4]Sogang University
[5]Independent Researcher

## ABSTRACT

As large language models (LLMs) are increasingly used across various applications, there is a growing need to control text generation to satisfy specific constraints or requirements. This raises a crucial question: *Is it possible to guarantee strict constraint satisfaction in generated outputs while preserving the distribution of the original model as much as possible?* We first define the *ideal distribution* — the one closest to the original model, which also always satisfies the expressed constraint — as the ultimate goal of guaranteed generation. We then state a fundamental limitation, namely that it is *impossible* to reach that goal through autoregressive training alone. This motivates the necessity of combining training-time and inference-time methods to enforce such guarantees. Based on this insight, we propose GUARD, a simple yet effective approach that combines an autoregressive proposal distribution with rejection sampling. Through GUARD's theoretical properties, we show how controlling the KL divergence between a specific proposal and the target ideal distribution simultaneously optimizes inference speed and distributional closeness. To validate these theoretical concepts, we conduct extensive experiments on two text generation settings with hard-to-satisfy constraints: a lexical constraint scenario and a sentiment reversal scenario. These experiments show that GUARD achieves perfect constraint satisfaction while almost preserving the ideal distribution with highly improved inference efficiency. GUARD provides a principled approach to enforcing strict guarantees for LLMs without compromising their generative capabilities.

## 1    INTRODUCTION

Large language models (LLMs) have demonstrated remarkable capabilities in generating human-like texts across a wide range of applications (OpenAI, 2023; Jiang et al., 2023; Yoo et al., 2024). Due to their usefulness, LLMs are being increasingly integrated into various downstream services and critical decision-making processes (Zelch et al., 2023; Arora & Arora, 2023; Kung et al., 2023). However, this widespread adoption raises concerns about both the reliability and the safety of LLM outputs, especially in high-stake scenarios where unintended behaviors could have significant consequences (Casper et al., 2023). Hence, it is crucial to address two key questions: *(1) How can we guarantee that all generated sequences from these powerful models meet specific constraints or requirements?* and *(2) How can we achieve this while preserving the original model's useful distribution as much as possible?* These two primary questions naturally lead to a third, practically important, question: *(3) How can we simultaneously obtain the two previous properties at a limited inference cost?*

Similar issues have been studied in the context of controlled text generation (Zhang et al., 2024a), where LLMs are conditioned on specific attributes to increase the likelihood of producing desired outputs. However, the associated methods do not provide the means to ensure that *all generated outputs strictly meet the desired constraints* — a problem we refer to as *guaranteed generation*. This

---

*Work done during an internship at NAVER Labs Europe.
†Correspondence to {`minbeomkim@snu.ac.kr, marc.dymetman@gmail.com`}

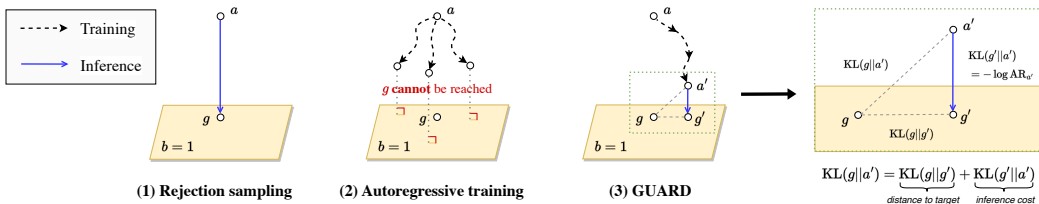

Figure 1: GUARD overview. Rejection sampling **(1)** can directly emulate $g$ from $a$, but it may incur a large inference cost when constraint $b$ is hard to satisfy. In the case of autoregressive training **(2)**, it is in general impossible for such a model to reach $g$ (see Theorem 1). GUARD **(3)**, on the other hand, first learns an approximation $a'$ of $g$ and then performs a simple form of rejection sampling using $a'$ as the proposal. This approach yields a distribution $g'$ which (i) strictly satisfies constraint $b$, (ii) minimizes inference cost, and (iii) is highly similar to $g$. Properties (ii) and (iii) are simultaneously enforced by minimizing $\text{KL}(g||a')$ as illustrated in the rightmost diagram (see Theorem 2).

is the gap we seek to address here. Our primary objective is then to study, in-depth, the characteristics of guaranteed generation within theoretically tractable scenarios. We attempt to clarify, both theoretically and experimentally, fundamental questions related to this critical topic, including how strict constraint enforcement affects the original properties of LLMs.

First, we formalize the concept of guaranteed generation over a base LLM by introducing the notion of a gold distribution, $g$, i.e., the ideal distribution we target. It is represented by a filtered energy-based model (EBM) (LeCun et al., 2006) that always satisfies the specified constraint $b$ — e.g., a binary filter function — while minimally diverging from the original distribution $a$ associated with the base LLM. Although we can compute the probability $g(y)$ for any arbitrary text $y$, this does *not* imply the existence of an autoregressive model $a'$ with the same distribution as $g$. We present an impossibility theorem (Theorem 1), demonstrating that in fact an exact match of this ideal distribution $g$ through an autoregressive model is generally impossible.

Still, as we will see, it is possible to train an autoregressive model to *approximate* $g$. However, finding such an approximation that *strictly* satisfies $b$ is far from obvious in both theory and practice. This insight motivates the use of inference-time methods such as Monte-Carlo sampling techniques. These methods exploit such approximations as proposal models and can strictly guarantee $b$, but they typically come with high inference costs that make their adoption impractical.

In order to address these issues, we propose GUARD,[1] a simple framework for guaranteed generation that combines approximation of the ideal distribution $g$ into a proposal distribution $a'$ with rejection sampling from the latter proposal. We establish a theorem (Theorem 2) about the relationship among the proposal $a'$, the GUARD output $g'$, and the ideal distribution $g$ through information geometry, by using the Pythagorean theorem for Information Projections (I-projections) (Csiszár & Shields, 2004). As shown in Figure 1, this theorem states how it is possible to optimize both inference speed at sampling time (by improving the acceptance rate) and distributional closeness while strictly satisfying guarantees, by optimizing the approximation of the proposal $a'$ towards $g$. We present several approaches for this approximation, stressing in particular the application to our situation of the Distributional Policy Gradient (DPG) technique (Parshakova et al., 2019; Khalifa et al., 2021; Korbak et al., 2022a;b; Go et al., 2023) for training a parametric model to approximate an arbitrary unnormalized target distribution represented by an EBM.

To validate these theoretical concepts, we conduct experiments in scenarios where the base LLM $a$ rarely satisfies the constraint $b$. We evaluate *how closely* our GUARD-based $g'$ approximates the gold distribution $g$ while maintaining a *high acceptance rate* in two scenarios with extensive analysis: a scenario with lexical constraints requiring specific strings to be included, and a sentiment reversal scenario with a positive ending constraint for stories with a negative opening. We demonstrate the effectiveness of GUARD in these scenarios and highlight the benefits of a novel warm-start variant of DPG, which involves initializing the training of $a'$ using constraint-aware prompting to bypass inefficient early stages of the proposal. Furthermore, we analyze how the original properties of the base model are degraded when the proposal diverges from the ideal distribution, thus connecting our theoretical results with empirical findings.

In summary, the main contributions of the paper are:

---

[1]The code is available at https://github.com/naver/guard.

1. The definition of the distribution $g$ as the ideal target for guaranteed generation; the proof that $g$ is, in general, unattainable by autoregressive models, and therefore by fine-tuning alone; the need to complement training-time methods with inference-time methods.

2. The proposal of the GUARD framework, combining a method for approximating $g$ by an autoregressive model $a'$ with a rejection sampler for enforcing the constraint; a proof that $\mathrm{KL}(g||a')$ controls both the efficiency of this sampler and its divergence relative to $g$.

3. Experiments with lexical and positive ending constraints, demonstrating that using GUARD leads to significantly improving the acceptance rate over rejection sampling from the base LLM $a$, while providing an excellent approximation to $g$; an analysis of the DPG and prompting approximation techniques, and the apparent limitations of prompting with respect to the diversity of $g$ in contrast to DPG; the warm-start combination of both techniques to speed up the approximation process.

## 2 FORMALIZATION OF GUARANTEED GENERATION

### 2.1 DEFINITION OF THE GOLD MODEL

Suppose that we have an autoregressive language model $a$ with vocabulary $V$, and let $a(y)$ be the probability of the output sequence $y \in \mathcal{Y} = V^*$.[2] Our goal is to transform $a$ into a model $p$ such that whenever $p$ generates $y$, then $y$ satisfies the *hard constraint* $b$, where $b$ is a binary function over $\mathcal{Y}$. Formally, this means that $p(y) > 0 \Rightarrow b(y) = 1$ for any $y \in \mathcal{Y}$. In other words, we want $p$ to *filter out* outputs that do not respect the constraint $b$.

Clearly, the above filtering requirement can be satisfied by many models; in particular, it would be trivially satisfied by any model generating only a single sequence $y$ fulfilling the constraint. So, additionally, we desire the target model $p$ to *minimally distort* the distribution associated with $a$. This criterion naturally leads to the following model $g$, which we will refer to as the *gold filtered model*, or simply *gold model*:

$$g(y) \propto a(y) \, b(y), \tag{1}$$

where $g(y)$ is the normalized distribution proportional to the value $a(y) \, b(y)$, in other words $g(y) = \frac{1}{Z} a(y) \, b(y)$, with $Z = \sum_{y \in \mathcal{Y}} a(y) \, b(y)$.

It is easy to see that $g$ is simply the distribution $a$ renormalized over the set of $y$'s that satisfy the constraint $b(y) = 1$. We can characterize $g$ equivalently as follows (see App. C.1 for discussion):

- $g$ is the distribution $a$ conditioned by the fact that $y$ satisfies $b$, i.e., $g(y) = a(y \mid [b(y) = 1])$.
- $g$ is the *I-projection* (Csiszár & Shields, 2004) of $a$ on the linear space $\mathcal{C}$ of distributions $p$ that respect the constraint everywhere, that is, such that $\forall y \in \mathcal{Y}, p(y) > 0 \Rightarrow b(y) = 1$ (in Figure 1, $\mathcal{C}$ was informally denoted by "$b = 1$"). In other words, $g$ is the distribution $p$ in $\mathcal{C}$ that minimizes the divergence $\mathrm{KL}(p||a)$.

### 2.2 FROM GOLD MODEL TO SAMPLER

While $g$ is a well-defined *distribution*, it is not immediately associated with a *sampler*, that is, to a procedure for generating samples following the distribution $g$. In particular, $g$ is not defined in an autoregressive form (which would directly result in a sampler), but in the form of an Energy-Based Model (LeCun et al., 2006), as the distribution corresponding to the normalization of the product $a(y) \, b(y)$.[3]

---

[2]An autoregressive model such as $a$ can be seen *both* as a *generator* of sequences $y \in \mathcal{Y}$ (that is, as a *sampler*) and as a probability *distribution* over $\mathcal{Y}$, where $a(y)$ is the probability of $y$. We sometimes use the notation $a$ both for the generator and for the distribution, when there is no risk of confusion. Note that many distributions, such as $g$ below, do not have this double nature: $g$ is not directly associated with a sampler.

[3]Formally, an EBM is any distribution $p$ presented in the form $p(y) \propto e^{-E(y)}$, where $E(y)$ is the "energy" of $y$. In other words, $p(y) = \frac{1}{Z} e^{-E(y)}$ where $Z$ is a normalization constant. Equivalently, $p$ can be presented in the form $p(y) \propto P(y)$ where $P$ is a nonnegative function of $y$ (also called a "potential"). In our specific case, this potential is equal to $a(y) \, b(y)$.

In order to associate a sampler to $g$, several approaches are possible. The first one consists of trying to find an autoregressive model (ARM) whose distribution is *identical* to $g$. Unfortunately, this is *impossible* in general, as we show below. Alternatively, an ARM $a'$ could be considered in order to provide a "good enough" approximation to $g$, but it is far from obvious how to ensure that such a sampler will both satisfy the constraint $b$ and remain close to the gold model. A third approach is to use the ARM $a'$ as a "proposal" inside some Monte-Carlo (MC) *inference-time* procedure. The framework that we propose, GUARD, is of this kind. It strictly guarantees the constraint and attempts to remain close to the gold model, while minimizing inference costs.

## 2.3 LIMITS OF AUTOREGRESSIVE MODELS FOR FILTERING PURPOSES

Let's consider the following simple formal example, to build some intuition. Let $a$ be a standard ARM and suppose that our constraint $b(y)$ is satisfied if and only if the sequence $y$ does *not* contain the token "bad". Is it possible to find an ARM $a'$ such that the distribution associated to $a'$ is equal to $g$? A first attempt could be to sample a large dataset of $y$'s from $a$, delete all $y$'s that do contain the token "bad", and fine-tune $a$ on the resulting dataset to obtain $a'$. However, while $a'$ would tend, to some extent, to avoid "bad" more often than $a$, it would not strictly guarantee this avoidance: standard fine-tuning methods never result in a softmax that is exactly zero on any token, including the token "bad", implying that the generation of an invalid output would remain possible.[4]

However, fine-tuning is not the only way to produce autoregressive models. Another approach would be as follows: we could, at each timestep $t$ of the generation of $y$, remove the token "bad" from the softmax vector $p(\cdot|y_{<t})$, and renormalize this vector to sum to 1. This would result in an ARM $a'$, strictly satisfying the constraint, but would actually cause $a'$ to have a distribution different from $g$, a phenomenon detailed in App. C.2.

So, obtaining an ARM with distribution $g$ is not obvious, even in very simple cases as above, and we discuss in App. C.2 why this is a widespread problem in practice. As we mention there, we are only aware of a few special cases where the problem is solvable, but, as we will now discuss, we do have a clear *negative* result: the problem has no general solution.

**A fundamental impossibility result**   We define a function from $V^*$ to $\mathbb{R}^d$ as *polynomial-time computable* (PTC) if and only if it can be computed in polynomial time relative to the length of its argument. A binary predicate is said to be PTC if it is a PTC function of its argument. An ARM $a$ is considered PTC if and only if the computation of the softmax vector at each step $t$ is a PTC function of the prefix up to that step.

The following result was inspired by the pioneering work of Lin et al. (2020b) on the general limitations of ARMs. See App. C.3 for a detailed statement and self-contained proof, adapted to the case of filtered models.

**Theorem 1.** *Under the assumption $P \neq NP$, there exists a PTC ARM $a$ and a PTC binary predicate $b$ such that no PTC ARM $a'$ has the same distribution as $g$.*

*Proof intuition.* We construct an $a$ that generates sequences encoding an instance of an NP-hard problem followed by a candidate solution to that problem, and a $b$ that checks (in polynomial time) the validity of the proposed solution. Then $g$ is a distribution whose support consists of all the sequences encoding a problem instance and a valid solution. If $a'$ were an ARM that corresponded to $g$, then it would be possible to check in polynomial time whether a given problem instance carried a non-zero probability mass, and therefore to decide the satisfiability of the problem instance, in contradiction to the generally accepted conjecture $P \neq NP$.

**Interpretation**: In practice, the PTC condition is satisfied by all standard ARM architectures, from Recurrent Neural Networks to Transformers of different flavors. Then, in essence, the theorem implies that it is in general impossible to fit $g$ with an ARM under such architectures.

---

[4]It should be noted more generally that because fine-tuning a model $a$ into $a'$ never puts a zero probability mass on any token, any sequence $y$ such that $a(y) > 0$ is still such that $a'(y) > 0$, that is, *fine-tuning can never fully eliminate any sequence.* This implies that all techniques relying solely on fine-tuning — e.g., PPO (Schulman et al., 2017) or DPO (Rafailov et al., 2023) — are limited in their ability to enforce strict constraints.

## 2.4 CLAIM: INFERENCE-TIME METHODS ARE NEEDED

While it can be difficult or even impossible to exactly fit $g$ with an ARM $a'$, it is still possible to *approximate* it with such a model and this can be done with various approaches such as (i) *training* $a'$ through Supervised Fine-Tuning (SFT) or DPG (Parshakova et al., 2019), or else (ii) through *prompting* $a$ with an instruction or few-shot examples attempting to express the constraint $b$ in natural language. While the resulting model $a'$ has the advantage that it can be used efficiently for generation, it typically does not strictly enforce $b$, our primary requirement, and we are not aware of general techniques that would allow an ARM $a'$ to strictly enforce $b$ while simultaneously limiting the distortion of $a'$ relative to $g$.

**Claim:** To achieve this objective, we posit that one has to rely on some *inference-time* Monte-Carlo (MC) algorithm that exploits the ARM approximation $a'$ as a proposal sampler but only retains samples satisfying the constraint $b$.

Such MC samplers can take different forms, from simple approaches such as Rejection Sampling to more involved MCMC techniques such as Metropolis-Hastings (Metropolis et al., 1953). The sampler used in GUARD, that we present now, is a basic form of rejection sampling, but with attractive properties compared to other approaches.

## 3 THE GUARD FRAMEWORK AND ITS PROPERTIES

Inside the GUARD framework, we assume that $a'$ is some autoregressive approximation to $g$. The GUARD sampler is then an elementary form of rejection sampling which can be described by the following algorithm: sample $y$ from $a'$ until $b(y) = 1$, then return this $y$ (**Algorithm 1**). We assume that there exists at least one $y \in \mathcal{Y}$ s.t. $a'(y) > 0$ and $b(y) = 1$. Such a procedure obviously guarantees that any sample $y$ satisfies the constraint, our fundamental requirement.

We remark that in the special case where $a' = a$, we are simply sampling from $a$ until we satisfy the constraint, and it is then easy to show that the sampler we obtain *has exactly the same distribution as $g$* (see App. C.4). In this case, the GUARD sampler meets two criteria: (1) it strictly satisfies the constraint, and (2) its associated distribution is close to $g$ — in fact, it is equal to $g$. However, the third important criterion, efficiency, falls short when $a$ rarely satisfies the constraint $b$, which is equivalent to $\text{KL}(g||a)$ being large (see Eq. 9 in App. C.5).

By approximating $g$ with some $a'$ that has a small $\text{KL}(g||a')$ relatively to $\text{KL}(g||a)$ — in other words, $a'$ is a better approximation to $g$ than $a$ — we can greatly improve the efficiency of GUARD, by sacrificing exact distributional match to $g$. Our core Theorem 2 below (see also the rightmost panel of Figure 1) characterizes this trade-off formally. Here, for a given $a'$, $g' = g'_{a'}$ denotes both the sampler resulting from Algorithm 1 and the associated distribution. $AR_{a'} \doteq \mathbb{E}_{y \sim a'} b(y)$ denotes the acceptance rate of the algorithm, that is, the probability that a sample from $a'$ respects constraint $b$ and thus is accepted (see App. C.5).

**Theorem 2.** *We have $KL(g||a') = KL(g||g') + KL(g'||a') = KL(g||g') - \log AR_{a'}$.*

*Proof sketch* (see App. C.6 for details). The first equality results from the fact that $g'$ is the I-projection of $a'$ on $\mathcal{C}$ and from the Pythagorean identity for I-projections (Csiszár & Shields, 2004). The second identity is obtained from a simple derivation showing that $\text{KL}(g'||a') = -\log Z' = -\log AR_{a'}$, where $Z'$ is the partition function $Z' \doteq \sum_{y \in \mathcal{Y}} a'(y) b(y)$.

**Interpretation**: $\text{KL}(g||a')$ is a measure of the divergence of $a'$ from $g$. As we saw earlier, it is typically impossible to narrow it to 0, but we can reduce it by approximation techniques. $\text{KL}(g||g')$ measures the divergence of the sampler $g'$ from the gold distribution $g$, and $-\log AR_{a'} \in [0, \infty)$ measures the "inefficiency" of the sampler, where $-\log AR_{a'} = 0$ corresponds to a maximum acceptance rate of 1. Because KL divergences are non-negative, $\text{KL}(g||a')$ is an upper bound of both $\text{KL}(g||g')$ and $-\log Z'$, which means that controlling the approximation quality of $a'$ relative to $g$ is the key to both controlling the quality of $g'$ relative to $g$ and also the efficiency of Algorithm 1.

**Approximating the target distribution** This last observation motivates the need to minimize $\text{KL}(g||a')$ to obtain a suitable $a'$. We consider several methods to achieve this. (1) Prompting can be used to encourage the output to respect the given constraint — a technique we refer to as

*constraint-aware prompting* (CAP). (2) Another approach consists in using Supervised Fine-Tuning (SFT). This involves sampling a large number of $y$'s from $a$, filtering them based on $b$ to create a dataset representing the $g$ distribution, and then fine-tuning $a$ on this dataset. However, when $AR_a$ is low, very few samples are retained, making this approach inefficient and difficult to use in practice, although it may still be useful for scientific analysis and comparisons. (3) The last method we consider is the *Distributional Policy Gradient (DPG)* technique for general EBMs (Parshakova et al., 2019; Korbak et al., 2022b; Go et al., 2023) here applied to the specific case of filtered models. This method samples $y$ from a proposal $a'$ initialized with $a$ and updates $a'$ by performing gradient descent on $\mathrm{KL}(g||a')$. It is important to mention that while both DPG and SFT seek to minimize $\mathrm{KL}(g||a')$ (see App. C.7 for more details), one key difference between them is that DPG is adaptive: as the proposal $a'$ approaches $g$, the frequency and quality of gradient updates increase, making it more efficient than SFT (Khalifa et al., 2021). While the original DPG technique shares SFT's initial low $AR_a$ during early training, we can improve this behavior by using constraint-aware prompting for the initial proposal — a method we refer to as "warm-start DPG" — to accept a rich amount of samples, thereby providing abundant gradient signals. Based on our experiments, this combination appears to be the most effective for guaranteed generation, and it is the one that we advocate. The full algorithm for warm-start DPG training in GUARD is provided in App. D as Algorithm 2.

**GUARD vs. MCMC and related sampling techniques**  Simple rejection sampling as in Algorithm 1 might at first sight seem naive relative to some more sophisticated techniques, such as Markov Chain Monte Carlo (MCMC) approaches. One such technique, Independent Metropolis-Hastings (IMH) (Robert & Casella, 2004), can use an autoregressive model as its proposal, and construct a random walk that converges to $g$ in the limit, while still guaranteeing a strict respect of the constraint. On inspection, however, each individual move of this random walk implies — and therefore is at least as costly as — a full run of Algorithm 1. We report in App. H experiments showing that IMH converges very slowly, making it an impractical approach, and motivating our choice of Algorithm 1 combined with a focus on finding a good approximator $a'$ to $g$. We also present in App. H some experiments based on Quasi-Rejection Sampling (QRS) (Eikema et al., 2022) — a technique also converging to $g$ in the limit — which, while still being more costly than Algorithm 1, is shown to present certain advantages over IMH.

## 4 Experiments

In this section, we describe the experiments conducted to evaluate the GUARD framework on simple guaranteed generation scenarios. As discussed in the previous section, we compare three approximation methods within GUARD: CAP, SFT, and DPG.[5] We have three main desiderata on GUARD's outputs: (1) strict adherence to constraint $b$, (2) similarity to the gold distribution $g$, and (3) limited inference cost.[6] Since $g'$ is filtered by rejection sampling in GUARD and thus satisfies the first criterion by definition, our main metrics in the experiments are the distributional closeness $\mathrm{KL}(g||g')$ and the inference cost defined by the acceptance rate $AR_{a'}$. Additional details about our metrics are provided in App. E.[7]

We are considering two scenarios, text generation under a lexical constraint (Section 4.1) and story generation under a positive ending constraint for sentiment reversal (Section 4.2), where samples from the base LLM $a$ rarely respect the constraint $b$ (i.e., $\mathrm{KL}(g||a) = -\log AR_a$ is high).[8]

---

[5]Training and evaluation were done using the `disco` toolkit: https://github.com/naver/disco.

[6]In terms of computation costs, reducing the *inference* cost related to the acceptance rate $AR_{a'}$ is our main focus, because this is the one involved in deployment. However, we also consider the *training* cost, in terms of the sampling budget required for training the approximation $a'$, which is independent of the inference cost.

[7]Note that in this work our evaluation focus is in terms of the closeness of $g'$ to the target gold model $g$ rather than in terms of the performance of $g'$ on downstream tasks like the ones proposed in benchmarks such as CommonGen (Lin et al., 2020a) — tasks that we see as directly influencing the design of the constraint $b$, and therefore the gold model $g$, which is *then* approximated with $g'$.

[8]If the acceptance rate of the base LLM $a$ is high enough, we can simply perform rejection sampling on proposal $a$ without any training. However, when violations of constraint $b$ are frequent, alternative methods are needed. This is the setting we explore in our experiments.

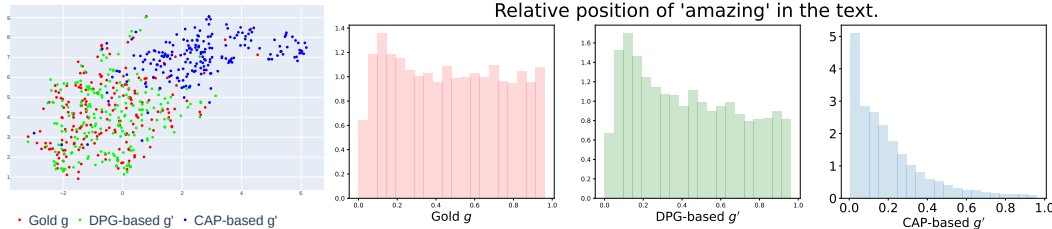

Figure 2: Left: UMAP visualization of the samples generated by different approaches in the lexical constraint scenario. Right: distribution of the relative position for the string "amazing" within the full sequences generated by each method. (Best viewed in color.)

## 4.1 UNCONDITIONAL GENERATION WITH LEXICAL CONSTRAINTS

**Task description** Our first task is to generate a text under a lexical constraint, i.e., such that a specific keyword is included in the generation. The LLM generates a text $y$ of 30 tokens, starting with the $\langle bos \rangle$ token. The lexical constraint $b(y)$ then either rejects or accepts this generated text based on whether it contains the string of the keyword. We use Gemma-2B (Riviere et al., 2024) as our base LLM $a$. We define the constraint $b$ as the binary function that is equal to 1 if and only if the generated text $y$ contains the string "amazing", following the experimental setup from Khalifa et al. (2021). This constraint has an acceptance rate $AR_a$ of approximately 0.0023 (i.e., on average, one sample is accepted every 435 samples drawn) under the original distribution $a$ with ancestral sampling and a temperature of 1.

**Empirical results** Figure 3 illustrates the learning curves of $KL(g||a')$ for various methods according to the sampling budget, i.e., the number of samples drawn from the proposal during training. SFT, which involves sampling $y$ from $g$, struggles due to the low $AR_a$, resulting in a limited supply of training data, relatively to the budget. Moreover, as training progresses, the increasing discrepancy between the policy $a'$ and the sampler $a$ (due to the off-policy nature of SFT) causes it to reach a plateau and stop improving after a certain point. In contrast, DPG shows steady and stable improvement, ultimately achieving a better $KL(g||a')$ than SFT. We explain this superior performance through two key factors: (1) *adaptive proposal* — the proposal distribution $a'$ in DPG adaptively becomes closer to the target distribution $g$ over time; and (2) *increasing $AR_{a'}$* — as the proposal $a'$ becomes better, the

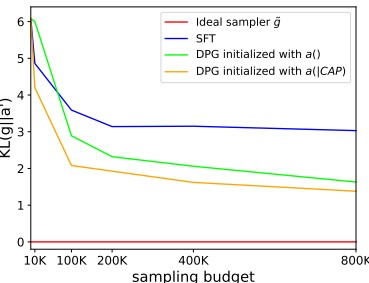

Figure 3: Evolution of $KL(g||a')$ as a function of the number of samples used for training with lexical constraints.

acceptance rate $AR_{a'}$ increases, providing more samples and thus a richer signal for estimating gradients. Additionally, warm-start DPG, i.e., initializing DPG with $a(\cdot|CAP)$ to leverage a constraint-aware prompt, encourages the model to generate "amazing", improves the initial acceptance rate, and allows us to avoid the slow early exploration stage where training samples are scarce. This results in consistently better performance throughout the whole training process.

Figure 4 depicts the results of the trade-off between $AR_{a'}$ and $KL(g||g')$ under a fixed sampling budget (800K). The point labeled as $\tilde{g}$ reflects the upper-bound performance associated to an ideal (but non-existent) sampler of the gold model, which would have no distributional difference with $g$ while achieving an acceptance rate of 1 (i.e., a $435\times$ improvement with respect to $a$). Both DPG and SFT show significantly improved acceptance rates in comparison to $a$. However, DPG consistently demonstrates superior distributional similarity, as measured by $KL(g||g')$. Interestingly, the CAP approach enforces a high acceptance rate without any training procedure, but it results in distributions that greatly diverge from the target $g$. Nevertheless, when DPG is initialized with $a(\cdot|CAP)$, it leverages the high initial $AR_{a'}$ to enable suffi-

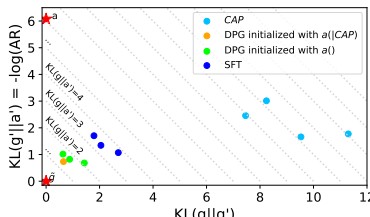

Figure 4: Results in terms of $KL(g||g')$ (i.e., distributional distance) and $-\log(AR_{a'})$ (i.e., inference cost) in the lexical constraint scenario. The dotted line indicates their sum, $KL(g||a')$, as stated in Theorem 2.

cient sampling. This, in turn, allows for a rapid reduction of $\text{KL}(g||g')$. As a result, the figure shows that this warm-start DPG strictly Pareto-dominates all SFT points. This means that for any given point representing an SFT result, warm-start DPG offers a better acceptance rate and a lower KL divergence simultaneously. In summary, the GUARD algorithm increases the acceptance rate from 0.0023 to 0.416 (almost $180\times$) compared to the base LLM $a$ while maintaining minimal distributional distortion.

**Analysis** This section analyzes the deeper phenomena underlying the $\text{KL}(g||g')$ results shown above. First, we visualize in Figure 2 the repartition of samples from the gold distribution $g$, DPG-based $g'$, and CAP-based $g'$ using UMAP (McInnes et al., 2018) after embedding them through DistilBERT (Sanh et al., 2019). DPG-based $g'$ and gold distribution $g$ overlap in similar regions, while the samples from CAP-based $g'$ cluster in a distinct area. This qualitative observation goes in line with the quantitative results from Figure 4, which reported a better $\text{KL}(g||g')$ for DPG-based $g'$ compared to CAP-based $g'$. Additionally, the CAP-based samples show reduced diversity (i.e., lower coverage). To quantify this loss of diversity, we measured Self-BLEU (Zhu et al., 2018), i.e., the $n$-gram similarity within the same sample group, and the average semantic similarity between sample embeddings.

As shown in Table 1, samples obtained from $g$ and DPG-based $g'$ exhibit similar diversity scores. However, samples filtered through constraint-aware prompting show a significantly narrower range of diversity. This reduced diversity is also evident in the positional bias analysis in Figure 2, which studies the relative position of the string "amazing" in the generated text. Samples from $g$ and DPG-based $g'$ show a relatively uniform distribution of "amazing" across token positions, whereas samples prompted to include the string "amazing" led to a heavy concentration of this word in early positions. Additional experimental details and analysis with other lexical constraints are provided in App. F, along with examples of generated samples.

| Approach | Self-BLEU ↓ | | | | Semantic similarity ↓ |
|---|---|---|---|---|---|
| | 2-gram | 3-gram | 4-gram | 5-gram | |
| Gold $g$ | 0.315 | 0.117 | 0.057 | 0.032 | 0.365 |
| DPG-based $g'$ | 0.342 | 0.124 | 0.067 | 0.040 | 0.372 |
| CAP-based $g'$ | 0.438 | 0.260 | 0.193 | 0.156 | 0.488 |

Table 1: Diversity of the samples generated by each approach in the lexical constraint scenario. Lower scores indicate a greater diversity.

## 4.2 CONDITIONAL GENERATION WITH A POSITIVE ENDING CONSTRAINT

**Task description** Next, we consider a more general guaranteed generation scenario with two factors: (1) conditional generation with prefixes, and (2) threshold-based constraints. The common usage of LLMs often involves conditional generation based on prefixes (e.g., when a prompt is used), and requirements are not always strictly binary by definition, as was the case for lexical constraints. They could instead be the result of a thresholded real-valued function (e.g., to distinguish between positive and negative sentiments). To reflect these two aspects, we investigate a story generation task with a positive ending constraint for sentiment reversal. In this task, given a negative story opening $X$, the LLM must generate a story continuation $Y$ and a last sentence $Z$ where the story ending $Z$ is highly positive (see Tables 8 and 9 for examples).

First, we prepared a positive sentiment scorer $b_{score}(y)$ (Hartmann, 2022) and defined our binary, threshold-based constraint as $b_\tau(y) = 1$ if and only if $b_{score}(y) > \tau$. We then selected our set of openings $X$ by collecting negative story openings from the ROCStories test set (Mostafazadeh et al., 2016), using only the first two sentences of stories for $X$ and imposing that $b_{score}(X) < 0.05$. To incorporate the domain knowledge of ROCStories, we use GPT-2 (Radford et al., 2019) fine-tuned on the ROCStories training set as our base LLM $a$. In this task, given a negative opening $X$, a completion $y = [Y; Z]$ is accepted if its final sentence $Z$ satisfies $b_{score}(Z) > 0.98$. The acceptance rate for the base LLM $a$ is $AR_a = 0.005$ (i.e., on average, one sample is accepted every 200 samples drawn). In other words, rejection sampling from $a$ approximately incurs a $200\times$ reduction in the acceptance rate compared to the ideal sampler $\tilde{g}$.

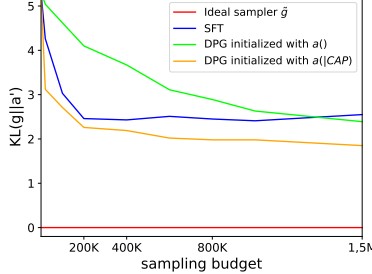

Figure 5: Evolution of $\text{KL}(g||a')$ as a function of the number of samples used for training in sentiment reversal.

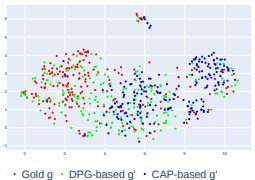
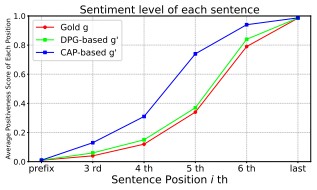

| Approach | Self-BLEU ↓ | | | | Semantic similarity ↓ |
|---|---|---|---|---|---|
| | 2-gram | 3-gram | 4-gram | 5-gram | |
| Gold $g$ | 0.744 | 0.569 | 0.458 | 0.337 | 0.876 |
| DPG-based $g'$ | 0.753 | 0.578 | 0.469 | 0.344 | 0.868 |
| CAP-based $g'$ | 0.827 | 0.701 | 0.621 | 0.554 | 0.956 |

Figure 6: Left: UMAP visualization of the samples generated by different approaches in the sentiment reversal scenario. Right: average positivity scores for each sentence position in the completions generated by each method. (Best viewed in color.)

Table 2: Diversity of the samples generated by each approach in the sentiment reversal scenario. Lower scores indicate a greater diversity.

**Empirical results** The sentiment reversal experiment shows similar results as the lexical constraint experiment. As illustrated in Figure 5, SFT reaches a plateau and stops improving after a certain point. In contrast, DPG continues to improve steadily as its proposal model adaptively approaches the gold model $g$, ultimately outperforming SFT for a 1.5M sampling budget. Nonetheless, using $a$ as the initial proposal in DPG results in inefficient learning due to rare gradient signals stemming from $a$'s very low acceptance rate. However, we observe that initializing DPG with the proposal $a(\cdot|\text{CAP})$ results in the best training efficiency. This technique bypasses the inefficient early exploration phase by leveraging a large amount of accepting samples thanks to prompting.

As shown in Figure 7, constraint-aware prompting improves the acceptance rate without any training, but results in a distribution that greatly diverges from $g$. Due to GPT-2 being a smaller model than Gemma-2B, the AR improvement from prompting is relatively less pronounced than in the lexical constraint experiment. In contrast, SFT and DPG, which optimize $\text{KL}(g||a')$, demonstrate an increase in AR while simultaneously showing less divergence from the target distribution. DPG, in particular, exhibits a distribution closer to the gold model $g$. In summary, the GUARD algorithm yields an AR improvement from 0.005 to 0.306 (almost $60\times$) compared to $a$ while preserving a high proximity to $g$.

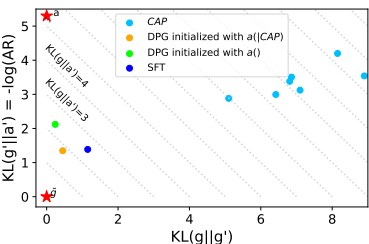

Figure 7: Results in terms of $\text{KL}(g||g')$ (i.e., distributional distance) and $-\log(AR_{a'})$ (i.e., inference cost) in the sentiment reversal scenario.

**Analysis** As with the lexical contraint experiments, we plot in Figure 6 a UMAP visualization of the DistilBERT-embedded samples obtained from $g$, DPG-based $g'$, and CAP-based $g'$. Similarly to these experiments, the samples from $g$ and DPG-based $g'$ overlap in the same regions, while the samples from the CAP-based $g'$ result in a more restricted coverage. This visualization confirms that while prompting can increase AR without sampling, it also leads to a reduction in distributional diversity. To quantitatively measure this phenomenon, we again calculated Self-BLEU scores and the average similarity between embeddings.

The results reported in Table 2 indicate that when conditioning on a specific prefix, the diversity is relatively lower compared to unconditional generation (as seen in Table 1). Nonetheless, samples from DPG-based $g'$ and $g$ exhibit similar diversity levels. Instead, samples obtained through CAP resulted in higher similarity, i.e., lower diversity. Figure 6 demonstrates that a positional bias is also observed in the case of a sentiment constraint. Compared to $g$ and DPG-based $g'$, when prompting to include a positive ending, the story continuation shows a more abrupt early shift towards positivity. This further confirms the existence of a positional bias in the constraint-aware prompting behavior — similarly to the lexical constraint scenario — and illustrates how this bias leads to distributional distortion. We provide additional details and results for this setting in App. G , including examples of generated samples.

## 5 RELATED WORK

**LLM alignment** LLM alignment techniques are widely regarded as the *de facto* method for integrating external desirable properties (e.g., helpfulness or harmlessness) into base language models (Kim et al., 2024). These techniques require access to a preference dataset, which consists of prompts, each associated with several completions, one of which is labeled as preferable based on

criteria like human judgment. Popular approaches in this family include RLHF (Ouyang et al., 2022), which trains a reward model from the preference dataset and fine-tunes the LLM through PPO (Schulman et al., 2017), and DPO (Rafailov et al., 2023), which combines these steps to directly fine-tune the model on the preference dataset. While these methods enable models to generate outputs that are better aligned with the implicit properties and values expressed in the preference dataset, such as harmlessness, there remains a gap between alignment and the goal of ensuring that all outputs are harmless. Alignment does not provide a straightforward means to enforce explicit constraints, as required in our guaranteed generation task. Consequently, despite their effectiveness, these alignment techniques, when used in a stand-alone manner (i.e., without additional inference-time processing), do not guarantee the satisfaction of a specific property or constraint in all generated outputs.

**Controlled text generation**   More closely related to our goal of guaranteed generation is the line of work on controlled text generation, which explicitly defines conditions or constraints (e.g., in the form of predicates) that outputs should satisfy. This topic gathers a large body of literature, and we thus let the reader refer to Zhang et al. (2024a) for a more comprehensive survey. Existing approaches include prompting to integrate the constraint (Zhang & Song, 2022; Yang et al., 2023), training techniques related to reinforcement learning (Khalifa et al., 2021; Korbak et al., 2022b; Go et al., 2023; Lu et al., 2022a) or learning local discriminators to encourage constraint satisfaction (Meng et al., 2022), or else relying on inference-time methods to impose the conditions without updating the original language model (Dathathri et al., 2019; Krause et al., 2020; Yang & Klein, 2021; Lu et al., 2022b; Liu et al., 2021; Eikema et al., 2022; Mireshghallah et al., 2022; Kim et al., 2023; Mudgal et al., 2023; Chakraborty et al., 2024).

The previously cited papers do not focus on strictly enforcing constraint satisfaction. However, two recent works (Zhang et al., 2023; 2024b) address this issue. Similarly to us, they are able to produce outputs that strictly satisfy the constraint, while maintaining proximity to the original model. They do so by approximating the LLM with a Hidden Markov Model (HMM), and by exploiting the dynamic programming properties of HMMs. This approach enables the computation of a weighted intersection of the HMM with a constraint over lexical items, resulting in a new HMM that always respects the constraint. Two key differences between their approach and ours are that (i) we handle arbitrary constraints — not necessarily of a lexical nature — which would be difficult to do through HMMs or similar finite-state mechanisms, and (ii) our focus is on producing an efficient sampler that is distributionally close to the gold target distribution $g$ that intrinsically represents the constraint $b$. In contrast, their focus is on producing a decoder that performs well on downstream tasks that have a looser relation to the constraint. We provide more details on these and some other related works in App. B.

## 6   CONCLUSION AND DISCUSSION

In this work, we formalize how to guarantee that LLMs perfectly meet specified requirements without compromising their usefulness. We first present a theoretical foundation addressing this challenge and state that it cannot be resolved using autoregressive models alone. To overcome this limitation, we introduce the GUARD framework, which approximates the gold distribution during training and exploits rejection sampling at inference time. To validate it empirically, we conduct experiments on two scenarios with hard-to-satisfy constraints, and confirm that outputs generated through GUARD provide guaranteed generation with significantly faster inference, while closely resembling the gold distribution.

The GUARD framework is potentially applicable to various crucial tasks that require strict compliance with specified requirements without diminishing the model's utility. For instance, in safety alignment tasks, GUARD could ensure that the model generates content in accordance with ethical guidelines and policy constraints, while achieving minimal loss of helpfulness. Similarly, in tackling the issue of *jailbreaking* — where users attempt to manipulate language models into producing prohibited or harmful content — GUARD could enforce compliance with safety protocols while maintaining the quality and informativeness of the responses. By integrating GUARD into these scenarios, we could effectively prevent the generation of undesirable outputs and enhance the reliability of LLMs. We believe that GUARD has the potential to improve both the safety and efficacy of language models across a wide range of applications.

ACKNOWLEDGMENTS

We would like to thank Laurent Besacier and Germán Kruszewski for their valuable feedback on this work.

MK acknowledges that this work was partially supported by an Institute of Information & communications Technology Planning & Evaluation (IITP) grant funded by the Korea government (MSIT) [No.RS-2022-II220184, Development and Study of AI Technologies to Inexpensively Conform to Evolving Policy on Ethics & NO.RS-2021-II211343, Artificial Intelligence Graduate School Program (Seoul National University)].

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

## A    LIMITATIONS

It is important to acknowledge the limitations of the GUARD framework, in particular when the filtering model $b$ fails to accurately assess the outputs relative to the true underlying requirements. In such cases, GUARD, while respecting $b$, could still produce undesirable outputs. The problem of designing the filter $b$ in such a way that it is able to accurately identify such outputs is undoubtedly important, but beyond the scope of the present work.

## B    COMPLEMENTS ON RELATED WORK

**Relation of our work to Zhang et al. (2023; 2024b)**    Similar to us, the approach from Zhang et al. (2023) is able to produce outputs that always satisfy the constraint, while maintaining proximity to the original model. It does so by approximating the LLM with an HMM, which, in contrast to the LLM, supports a dynamic programming approach to complying with lexical constraints over the output, and therefore the ability to weigh the long-term consequences of local next-token choices relative to these constraints. When the HMM approximation to the LLM is good enough (which depends in particular on the number of states of the HMM), the same HMM can be used at inference time with different lexical constraints without need of retraining.

While such an approach is relevant for guaranteed generation, we note several fundamental differences with what we do:

1. We handle *arbitrary logical constraints* (that is, binary predicates), not necessarily of a lexical nature, which would be difficult to handle through HMMs or similar finite-state mechanisms.

2. We give a central status to the "gold" constrained distribution $g$, which is the distribution that minimally deviates from the original distribution while still fully satisfying the constraint, and we evaluate the quality of our sampler in terms of distributional distance to this distribution $g$. Zhang et al. (2023) do not focus on the evaluation of a *sampler* relative to a reference distribution, but rather on the evaluation of a *decoder* in terms of downstream tasks which have a looser relation with the constraint.

3. We study the trade-off between the quality of the sampler and its efficiency in the context of a simple rejection sampling mechanism based on an autoregressive approximation $a'$ of $g$ and show that quality and efficiency are both directly controlled by the divergence $\mathrm{KL}(g||a')$. This in turns motivates our interest in an approximation technique that focuses on minimizing this divergence, such as DPG.

The work from Zhang et al. (2023) is extended in a follow-up (Zhang et al., 2024b) which considers lexical constraints defined through deterministic finite automata. As a side observation, in App. C.2 (*Solvable instances of the problem*), we have briefly mentioned that in case the base ARM and the lexical constraint are based on weighted finite state automata, the intersection of these automata could be constructed and sampled from. We note that such weighted automata are naturally very much related to HMMs, and might be interesting to study in their own right.

**NADO sampler (Meng et al., 2022)**    The NADO distribution introduced in Meng et al. (2022) is trained with an objective similar to ours — namely, it tries to approximate a distribution comparable to our gold filtered distribution $g$. As a training method to obtain an autoregressive distribution similar to our $a'$, it has some analogies with (i) SFT, by training on samples from the filtered distribution, and (ii) DPG, by also performing a kind of distributional matching, but with more emphasis on local (i.e., token-level) decisions. Contrary to DPG, NADO does not directly have the objective of minimizing the divergence $\mathrm{KL}(g||a')$, which is the determining factor for the success of GUARD as shown in Theorem 2. Using NADO for $a'$ would nonetheless be interesting as a follow-up work, to see whether the value of its divergence is competitive with the cases we have explored, leading to improved performance for GUARD.

## C  ADDITIONAL BACKGROUND AND THEORETICAL DETAILS

### C.1  CHARACTERIZATION OF $g$

We have defined the distribution $g$ through the formula $g(y) = \frac{1}{Z} a(y) b(y)$, with $Z = \sum_{y \in \mathcal{Y}} a(y) b(y)$. It is straightforward to see that $g$ is simply the distribution of $a$ renormalized over the subset $\mathcal{Y}_1 \doteq \{y \in \mathcal{Y} : b(y) = 1\}$ and equivalently that $g$ is the distribution of $a$ conditioned on $b(y) = 1$, in other words $g(y) = a(y \mid [b(y) = 1])$.

A less obvious observation is the fact that $g$ is also the I-projection (Csiszár & Shields, 2004, Section 3) of $a$ on the space $\mathcal{C}$ of all distributions over $\mathcal{Y}$ that satisfy the constraint $b(y) = 1$ everywhere.[9] Thus, according to the definition of I-projection, $g$ is the distribution $p$ in this space that minimizes the divergence $\mathrm{KL}(p||a)$. *In other words, $g$ is, among the distributions that fully satisfy the constraint, the one that minimally distorts $a$, in terms of KL divergence* — a fact that further supports our characterization of $g$ as the ideal guaranteed model associated with $a$ and $b$.

Let's provide a self-contained proof of that fact. We have:

$$\mathrm{KL}(g||a) = \mathbb{E}_{y \sim g} \log \frac{g(y)}{a(y)} = \mathbb{E}_{y \sim g} \log \frac{a(y)b(y)}{Z\,a(y)} = -\log Z + \mathbb{E}_{y \sim g} \log b(y) = -\log Z, \qquad (2)$$

$$\mathrm{KL}(p||g) = \mathbb{E}_{y \sim p} \log \frac{p(y)}{g(y)} = \mathbb{E}_{y \sim p} \log \frac{Z\,p(y)}{a(y)b(y)} = \log Z + \mathbb{E}_{y \sim p} \log \frac{p(y)}{a(y)} = \log Z + \mathrm{KL}(p||a). \tag{3}$$

Hence we have $\mathrm{KL}(p||a) = \mathrm{KL}(p||g) + \mathrm{KL}(g||a)$, and because KL divergences are nonnegative, we see that $\mathrm{KL}(g||a) \leq \mathrm{KL}(p||a)$ which proves that $g$ is the unique[10] distribution $p$ in $\mathcal{C}$ that minimizes $\mathrm{KL}(p||a)$, namely, that $g$ is the I-projection of $a$ on $\mathcal{C}$.

**Controlling the distortion of $a$'s capabilities through the use of $g$**   The fact that $g(y) = \frac{1}{Z} a(y) b(y)$ implies that for any two sequences $y, y'$ with $b(y) = b(y') = 1$ and $a(y) > 0$, we have $g(y) > 0$ and:

$$\frac{g(y')}{g(y)} = \frac{a(y')}{a(y)}.$$

In words, this means that if $y'$ is $\alpha$ times more likely than $y$ relative to the original model $a$, and if both $y$ and $y'$ respect the constraint, than $y'$ is still exactly $\alpha$ times more likely than $y$ relative to $g$.

For example, suppose that, relative to $a$, the sentence "Potatoes were first cultivated in South America" is 100 times more likely than the sentence "Potatoes were first cultivated in Africa", and that both sentences satisfy the constraint, then the first sentence will still be 100 times more likely than the second relative to $g$.

So $g$ does not *distort* the knowledge/reasoning capabilities of $a$ over outputs that respect the constraint. *This property is not true for alternative models*, even if they only produce outputs respecting the constraint, and is a clear advantage of $g$ as a target model.

### C.2  ON THE LIMITATIONS OF AUTOREGRESSIVE MODELS IN REPRESENTING $g$

We start the discussion with two simple examples illustrating the difficulty of representing $g$ with an autoregressive model.

**Avoidance example**   First, we come back to the situation described in the main text where the constraint $b$ prohibits the token "bad" to appear in $y$. We saw that fine-tuning could not completely guarantee this constraint, because it would never put zero mass on the token "bad". Furthermore, fine-tuning would be pretty inefficient, even as an approximator, for this "avoidance" situation:

---

[9]$\mathcal{C}$ can also be seen, in the terminology of (Csiszár & Shields, 2004), as the linear variety of distributions $p$ that satisfy the expectation constraint (also called moment constraint) $\mathbb{E}_{y \sim p} b(y) = 1$, because here $b$ is a function with values in $\{0, 1\}$, and therefore such $p$'s are identical with $p$'s that satisfy the constraint on all $y$'s.

[10]Uniqueness comes from the fact that any $p$ such that $\mathrm{KL}(g||a) = \mathrm{KL}(p||a)$ is also such that $\mathrm{KL}(p||g) = 0$ — because of the identity above — and therefore is such that $p = g$.

a huge dataset would probably be necessary for significantly reducing the probability of "bad", because the fine-tuning process would only see examples not containing "bad", with no explicit negative examples. As mentioned in the text, another approach, not involving fine-tuning, would be, at each timestep $t$ during the generation of $y$, to remove the token "bad" from the softmax vector $p(\cdot|y_{<t})$, and renormalize this vector to sum to 1.[11] This would result in an autoregressive model $a'$ that always respects the constraint, but whose distribution would differ from $g$. Intuitively, this would be because the distribution for $g$, which corresponds to natural sequences from $a$ not containing "bad", would anticipate early on, for the initial tokens of $y$, the fact that "bad" cannot appear later, while $a'$ would ignore any such influence.[12]

**Enforcement example**    Conversely, rather than trying to avoid a certain token such as "bad", we could try to *enforce* the presence of a token such as "good" somewhere in the sequence $y$. In other words, we would have $b(y) = 1$ iff $y$ contains the token "good". Here, we could again sample a large dataset from $a$, keep only the sequences containing "good", and fine-tune an ARM $a'$ on the resulting dataset. However, while $a'$ would be better than $a$ at producing sequences containing "good", it could still produce other sequences, and therefore would not strictly guarantee the constraint (although this fine-tuning would probably be more effective as an approximator than in the previous example of avoiding "bad", because at least it could rely on instances explicitly containing the token). Again, we could try another autoregressive approach: we could, at each step of the generation of $y$, check whether we have already produced the token "good", and if not, continue the generation of $y$ (i.e., prevent the generation of an $\langle eos \rangle$ token). This would result in an ARM $a'$, which would guarantee strict satisfaction of the constraint, but the distribution of the sequences produced by $a'$ would be very different from $g$. This is because, similarly to the previous case, while $g$ would anticipate in its distribution over early tokens the necessity to produce "good" at a later stage, $a'$ would totally ignore such influence of "good" over the early tokens. We intuitively expect that $a'$ would tend to produce longer sequences than $g$, because there would be little incentive to produce "good" early on during the generation.[13] This was verified through a simple experiment in App. F (*Comparison to a heuristic sampler*), which confirmed that such an approach does not provide a satisfactory approximation to $g$.

In the two previous examples, of course, we did not provide a *proof* that modeling $g$ with an autoregressive model was impossible, but what we can say is that we are unaware of a technique for doing so, in both cases, and are skeptical about its existence.

**Solvable instances of the problem**    For these two examples, the situation with standard LLMs such as $a$, built over neural networks, contrasts strongly with what would be the case if $a$ were an $n$-gram language model. In that case, $a$ would be equivalent to a probabilistic weighted finite-state automaton (WFSA) (Mohri, 1997), a classical form of autoregressive model, and such constraints as the presence or absence of a specific token could be realized through intersection between such automata, resulting in another probabilistic automaton, with the same distribution as $g$. However, such intersections, which rely on dynamic programming over the finite set of states of these models, are unavailable in the domain of neural LMs.[14]

Thus, to summarize, if $a$ were an $n$-gram language model, where $b$ was a constraint similar to those considered in the two examples above, the problem of modeling $g$ through an ARM would be

---

[11]This is a special case of a more general technique based on lexical constraints described by finite-state automata, see e.g. Willard & Louf (2023).

[12]The intuition may be clearer with a more situation-specific word such as "malaria": in order to eliminate sequences containing this word from $a$, $g$ has to depreciate the probability of prefixes that naturally lead to that word — such as "This proliferation of mosquitoes raises the risk of" — while an $a'$ intervening on the local generation of "malaria" would not perform such anticipation.

[13]Again, the intuition might be clearer if the constraint was about producing a rarer word than "good", associated with a specific situation, for instance "trophy". Sequences from $a$ containing this word, which would be selected by $g$, would tend to start differently than generic sequences from $a$, but $a'$ would not show this kind of influence.

[14]Zhang et al. (2024b) recently proposed a technique with some similarities to what we are describing here, by performing the intersection of an HMM (a formalism with strong connections with WFSAs) with a finite-state automaton imposing lexical constraints. In order to apply this technique to a standard neural LLM, the authors propose to first approximate this LLM with an HMM, before applying the finite-state constraints. See Section 5 and App. B for more details on that work.

*solvable*. Another instance where this problem is solvable — staying this time in the familiar domain of neural language models — is when $a$ is a standard LLM, and $b$ is the constraint which imposes the presence of a certain *prefix* $[y_1; \ldots; y_t]$ at the beginning of the generated sequence $y$. It is easy to see that the generative process that samples $y_{t+1} \sim a(\cdot | [y_1; \ldots; y_t])$, $y_{t+2} \sim a(\cdot | [y_1; \ldots; y_{t+1}])$, and so on, in the standard way, corresponds to an autoregressive model $a'$ that has the same distribution as $g$. However, again, this is a very special case, and as soon as the constraint does not refer to such an initial section of $y$, the problem becomes much more challenging.

**No general solution exists**  To conclude, we see that in practice, even in quite simple cases, it seems challenging to model $g$ with an ARM. Theorem 1 actually proves that the general problem is unsolvable, by relying on a more abstract mathematical situation.

## C.3  DETAILS ABOUT THEOREM 1

A more precise statement of Theorem 1 is provided below:

**Theorem 1'.** *Under the assumption $P \neq NP$, there exists a polynomial-time computable (PTC) ARM $a$ and a PTC binary predicate $b$ such that no PTC ARM $a'$ has the same support[15] as the gold model $g$, and in particular, has the same distribution as $g$.*

In this statement, when we speak of an "autoregressive model", we mean a model $m$ over sequences $y \in V^*$, with $V$ a finite vocabulary of tokens, which generates $y$ in the following way:

1. At each time-step $t$, given the prefix $y_{<t} = [y_1; \ldots; y_{t-1}]$ of already generated tokens, a function $f$ is applied on the prefix, where $f(y_{<t}) \in \mathbb{R}^{|V|}$. In other words, the value of the function is a vector of real numbers over the vocabulary $V$.

2. A softmax transformation is applied to the vector $f(y_{<t})$, yielding a probability distribution over $V$.

3. The token $y_t$ is sampled from this distribution, and the generation proceeds recursively.

As for the PTC qualification, we say that a function is PTC iff it can be computed in polynomial time relative to the length of its argument. In particular, the function $f$ is PTC iff it is computable in polynomial time relative to the length $t-1$ of the prefix $y_{<t}$, and the binary predicate $b$ is PTC iff it is computable in polynomial time relative to the length of $y$.

The PTC assumption about $f$ is essential here. It is satisfied by all standard implementations of autoregressive models, from RNNs to Transformers of different flavors. As for the PTC assumption about $b$, without it, the theorem would be devoid of interest.

### C.3.1  PROOF SKETCH

Consider any NP-complete problem, for instance 3SAT (Karp, 2010), a problem also considered by (Lin et al., 2020b). Now consider a textual sequence $y$ of the following form: `problem#assignment`, where the first part is some encoding of a 3SAT problem instance over $k$ boolean variables, and the second part is the encoding of a candidate assignment (i.e. 0 or 1) for each of the $k$ variables.

Two examples of such sequence encodings would be:[16]

$$x1 \lor \neg x2 \lor x3 \,; \neg x1 \lor \neg x3 \,\#\, 1\,1\,0$$
$$x1 \lor \neg x2 \lor x3 \,; \neg x1 \lor \neg x3 \,\#\, 0\,1\,0$$

where the candidate assignment in the first example satisfies the problem instance (is "valid"), while it does not in the second example (is "invalid").

As is obvious, it is possible to check the validity of the assignment relative to the problem instance in polynomial time relative to the length of $y$. On the other hand, under the standard conjecture

---

[15]The support of a distribution $p$ over $\mathcal{Y}$ is the set of $y$'s such that $p(y) > 0$.

[16]Here the token vocabulary would be $V = \{x, 0, 1, 2, 3, 4, 5, 6, 7, 8, 9, \lor, \neg, ;, \#\}$.

that $P \neq NP$, it is impossible, by looking at the `problem` instance, to check in polynomial time whether this `problem` instance is satisfiable. In other words, it is impossible to check whether there exists an `assignment` that is valid relative to this problem instance.

Now consider a PTC autoregressive model $a$ that is able to generate, possibly among other sequences, *all* sequences that cover any syntactically well-formed `problem#assignment`, even if the `assignment` is only a possible candidate, but does not actually satisfy the `problem` instance.[17] Whatever the exact nature of $a$, the key point is that, for any sequence $y$ encoding a well-formatted `problem#assignment`, we have $a(y) > 0$. Crucially, this is true whether or not the `assignment` is actually *valid* relative to the `problem`.

Given this $a$, now consider a predicate $b(y)$ that checks two aspects, a syntactic one and a semantic one:

1. Syntactic check: whether $y$ respects the format `problem#assignment`, and if not, rejects $y$.

2. Semantic check: assuming that the first check was successful, $b$ now checks that the `assignment` satisfies the `problem`.

Both checks can be done in polynomial time, as could be easily shown by being more precise on the exact nature of the encoding done, and by constructing small programs that would perform both checks. We do not go into the details here, that are both lengthy and straightforward.

Based on these $a$ and $b$, $g(y)$ is a distribution that gives positive probability to any $y$ of the form `problem#assignment`, where the `assignment` satisfies the `problem`, and a zero probability to any other $y$. Thus the support of $g$ is the set of `problem#assignment` $y$'s that are valid.

Now suppose that there exists some PTC autoregressive model $a'$ such that the support of $a'$ is equal to the support of $g$. Then $y$ is generated by $a'$ iff $y$ is of the form `problem#assignment`, where `assignment` satisfies `problem`. In particular, this means that $a'$ starts by generating a `problem` that is satisfiable, and furthermore, that it can only generate satisfiable problems. Now, this provides us with a decision procedure to determine whether a given `problem` is satisfiable or not, namely, we give the `problem` as a prefix to $a'$, and ask $a'$ whether it gives a zero probability or a positive probability to this prefix. This can be done by running $a'$ autoregressively over this prefix.

The time complexity of that decision procedure can be computed as follows. Let us write the prefix $pref$ in the form $pref = [x_1; \ldots; x_{pl}]$, with $pl$ denoting the prefix length. Then we have $a'(pref) = a'(x_1) \cdot a'(x_2|x_1) \cdot \ldots \cdot a'(x_{pl}|[x_1; \ldots; x_{pl-1}])$, where at each step $a'(x_t|[x_1; \ldots; x_{t-1}])$ we invoke the autoregressive model to compute the probability of $x_t$ given the previous tokens. This computation occurs in polynomial time relative to $t$. Therefore, the whole autoregressive computation of $a'(pref)$ is performed in polynomial time relative to the length of the prefix $pl$.

Overall, we now have a polynomial time algorithm that decides whether the `problem`, as encoded by $pref$ is satisfiable. This contradicts the $P \neq NP$ conjecture, and we conclude that the support of $a'$ is different from the support of $g$, and in particular that, as distributions, $a' \neq g$, which ends the proof.

**Note.** We believe both the statement of the theorem and its proof have some novelty relative to previous discussions of the limitations of autoregressive models. First, concerning the statement, our focus is on filtered autoregressive models, that is, conceptually very simple modifications of a base model. Indeed, we are just speaking of a model $a$ being conditioned by a predicate $b$. This may allow a clearer intuition of what is really going on. Second, concerning the proof, we take care (1) to avoid a delicate (albeit possible) autoregressive characterization of syntactically well-formed $y$'s of the form `problem#assignment`, by delegating the syntactic verification to $b$, an obviously polynomial-time check; and, more importantly, (2) to avoid looking for a proof relying on the

---

[17]Such an $a$ could be trivial, for instance it could generate all sequences of finite length over a vocabulary of tokens relevant for encoding the problem instance and the candidate assignment, even those that do not respect the proposed format, but it could also easily be much more focused on generating sequences of the proper syntactic format, for instance by being fine-tuned on such sequences. However, we do not want to *require* that $a$ *only* generates sequences of the proper syntactic format, because, although it is the case that this requirement can be fulfilled by certain autoregressive models, it requires more sophisticated machinery than what we actually need for our proof.

impossibility of finding in polynomial time a valid assignment for a *satisfiable* problem instance, because this would involve a preliminary distinction between satisfiable and unsatisfiable instances, something that cannot be done in polynomial time. Our proof avoids this issue completely, by focusing on full `problem#assignment` pairs, of which the validity *is* easily checked in polynomial time.

### C.4   CHARACTERIZATION OF $g'$

Let $P'(y) \doteq a'(y)b(y)$ and $p'(y)$ be the distribution such that $p'(y) \propto P'(y)$. Then $p'(y) = \frac{1}{Z'}P'(y)$ with $Z' \doteq \sum_{y \in \mathcal{Y}} P'(y)$. We have $P'(y) \leq a'(y) \cdot 1$, and $r(y) \doteq \frac{P'(y)}{a'(y) \cdot 1} = b(y)$. According to the fundamental result about rejection sampling, see (Liu, 2004, Section 2.2), $p'$ is the distribution obtained by sampling $y \sim a'$ and accepting $y$ with probability $r(y)$, which in our case is either 0 or 1. This is exactly what Algorithm 1 does, and therefore $g'$ has the same distribution as $p'$, in other words:

$$g'(y) = \frac{1}{Z'}a'(y)b(y), \tag{4}$$
$$\propto a'(y)b(y). \tag{5}$$

We finally note that when $a' = a$, Eq. 5 implies that the distribution $g'$ associated with Algorithm 1 is actually equal to the gold model $g$.

### C.5   RELATIONSHIP BETWEEN $\mathrm{KL}(g'||a')$ AND ACCEPTANCE RATE $AR_{a'}$

By the same reasoning as in App. C.4, applied to $a'$, we see that $g'(y) \propto a'(y)b(y)$, and therefore $g'(y) = \frac{1}{Z'}a'(y)b(y)$ where $Z' = \sum_{y \in \mathcal{Y}} a'(y)b(y)$.

It is easy to see that $Z'$ is also the probability that a sample from $a'$ is accepted by Algorithm 1, namely the acceptance rate $AR_{a'}$, and therefore $AR_{a'} = Z'$.

We then have $\mathrm{KL}(g'||a') = \mathbb{E}_{y \sim g'} \log \frac{g'(y)}{a'(y)}$, hence:

$$\mathrm{KL}(g'||a') = \mathbb{E}_{y \sim g'} \log \frac{b(y)}{Z'} = \mathbb{E}_{y \sim g'} \log \frac{1}{Z'}$$
$$= -\log Z' \tag{6}$$
$$= -\log AR_{a'} . \tag{7}$$

Because $a$ and $g$ are special cases of $a'$ and $g'$ respectively, we also have:

$$\mathrm{KL}(g||a) = -\log Z \tag{8}$$
$$= -\log AR_a . \tag{9}$$

### C.6   CORE THEOREM ABOUT GUARD

The fact that

$$\mathrm{KL}(g||a') = \mathrm{KL}(g||g') + \mathrm{KL}(g'||a')$$

is a consequence of the *Pythagorean Identity* for I-projections, as stated in Theorem 3.2 of Csiszár & Shields (2004). In order to apply the theorem, we observe that the set $\mathcal{C}$ of distributions $p$ such that $p(y) > 0 \Rightarrow b(y) = 1$ is identical to the linear family of distributions that respect the expectation constraint $\mathbb{E}_{y \sim p} b(y) = 1$, because in our case $b(y)$ can only take the values 0 or 1. The distribution $g'$ is the I-projection of $a'$ onto $\mathcal{C}$, and according to the theorem, for any distribution $p \in \mathcal{C}$, we have $\mathrm{KL}(p||a') = \mathrm{KL}(p||g') + \mathrm{KL}(g'||a')$. Because $g$ is in $\mathcal{C}$, we can conclude (see Figure 8).

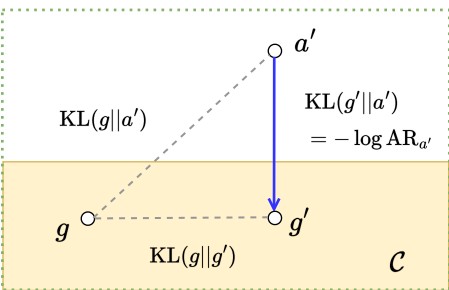

Figure 8: (Same as right panel of Figure 1.) The $g'$ distribution is the I-projection of $a'$ onto the space $\mathcal{C}$ of distributions respecting the constraint, that is, $g'$ is the distribution $p$ in $\mathcal{C}$ that minimizes $\text{KL}(p||a')$. The Pythagorean theorem says that, for any such distribution $p$, $\text{KL}(p||a') = \text{KL}(p||g') + \text{KL}(g'||a')$, in particular $\text{KL}(g||a') = \text{KL}(g||g') + \text{KL}(g'||a')$. Independently of this fact, we also have $\text{KL}(g'||a') = -\log AR_{a'}$, where $AR_{a'}$ is the acceptance rate associated with Algorithm 1.

We can also provide a simple, direct proof of this equality for our specific case. We have:

$$\text{KL}(g||g') = \mathbb{E}_{y \sim g} \log \frac{g(y)}{g'(y)} = \mathbb{E}_{y \sim g} \log \frac{Z'g(y)}{a'(y)b(y)}$$

$$= \log Z' + \mathbb{E}_{y \sim g} \log \frac{g(y)}{a'(y)} = \log Z' + \text{KL}(g||a').$$

$$\text{KL}(g'||a') = \mathbb{E}_{y \sim g'} \log \frac{g'(y)}{a'(y)} = \mathbb{E}_{y \sim g'} \log \frac{a'(y)b(y)}{Z'a'(y)}$$

$$= -\log Z'.$$

Hence $\text{KL}(g||a') = \text{KL}(g||g') + \text{KL}(g'||a')$.

We then apply the identity from Eq. (7) and conclude the proof.

### C.7 MINIMIZATION OF $\text{KL}(g||a')$

As discussed in Section 3 and in the interpretation of Theorem 2, minimizing $\text{KL}(g||a')$ results in finding a suitable $a'$ for GUARD. In this section, we show that both the SFT and DPG approximation techniques seek to minimize this objective.

The SFT loss corresponds to the cross-entropy of the model $a'$ to be learned, using samples from $g$, i.e., $-\mathbb{E}_{y \sim g} \log a'(y)$. Minimizing this term with respect to $a'$ is equivalent to minimizing $\text{KL}(g||a')$ since $\text{KL}(g||a') = \mathbb{E}_{y \sim g} \log \frac{g(y)}{a'(y)} = \mathbb{E}_{y \sim g} \log g(y) - \mathbb{E}_{y \sim g} \log a'(y) = H_g - \mathbb{E}_{y \sim g} \log a'(y)$ where $H_g$ is the entropy of $g$, which is a constant independent of $a'$.

The objective of DPG is also to minimize the KL divergence (or equivalently, the cross-entropy) between the target distribution $g$ and the learned policy $a' = \pi_\theta$ as detailed in Parshakova et al. (2019, Section 3.2). The derivation can be summarized as follows: $\nabla_\theta \text{KL}(g||\pi_\theta) = \nabla_\theta(H_g - \mathbb{E}_{y \sim g} \log \pi_\theta(y)) = -\mathbb{E}_{y \sim g} \nabla_\theta \log \pi_\theta(y) = -\mathbb{E}_{y \sim \pi_\theta} \frac{g(y)}{\pi_\theta(y)} \nabla_\theta \log \pi_\theta(y)$, where the last step is obtained by applying importance sampling using $\pi_\theta$ as the proposal. Performing gradient descent on $-\mathbb{E}_{y \sim \pi_\theta} \frac{g(y)}{\pi_\theta(y)} \nabla_\theta \log \pi_\theta(y)$ leads to the DPG update rule shown at Line 21 of Algorithm 2 (see below).

## D GUARD'S TRAINING ALGORITHM

Algorithm 2 illustrates the GUARD training process with DPG. The optional orange-colored steps — which correspond to what we are referring to as the warm-start initialization of DPG — focus on increasing the $AR_{\pi_\theta}$ through the CAP proposal, while the remaining steps follow the version of the DPG algorithm presented in Go et al. (2023, Algorithm 1). By initializing DPG training with

---

**Algorithm 2** GUARD training with warm-start DPG

---

1: **Input:** $a(\cdot)$, $b(\cdot)$, CAP, total sampling budget $BG$, CAP sampling budget $bg$
2: $\pi_\theta \leftarrow a(\cdot)$                ▷ Policy
3:
4: ▷ Initialization with CAP proposal
5: $D \leftarrow \emptyset$             ▷ Dataset of CAP samples
6: Sample $\{y_1, \ldots, y_i, \ldots, y_{bg}\}$ from $a(\cdot|\text{CAP})$
7: **for** $i = 1$ to $bg$ **do**
8:     **if** $b(y_i) = 1$ **then**
9:         Append $y_i$ to $D$
10: Fine-tune policy $\pi_\theta$ on dataset $D$
11:
12: ▷ Training with $\pi_\theta$ proposal
13: $P(y) \leftarrow a(y)b(y)$             ▷ Unnormalized target
14: $Z \leftarrow 0$              ▷ Partition function
15: $N \leftarrow 0$              ▷ Sample count
16: **while** $N < BG - bg$ **do**       ▷ Remaining sampling budget $BG - bg$
17:     Sample $y$ from $\pi_\theta$
18:     $N \leftarrow N + 1$
19:     $Z \leftarrow \frac{(N-1)Z + \frac{a(y)b(y)}{\pi_\theta(y)}}{N}$       ▷ Moving average estimate of $Z$
20:     $p(y) \leftarrow P(y)/Z$
21:     $\theta \leftarrow \theta + \alpha \frac{p(y)}{\pi_\theta(y)} \nabla_\theta \log \pi_\theta(y)$       ▷ Core DPG update rule
22: **Output:** $\pi_\theta$

---

a high-AR $\pi_\theta$ obtained through CAP, we can supply a large number of samples to the $\text{KL}(g||a')$ minimization process, leading to more efficient training.

# E    ADDITIONAL METRICS DETAILS

**Acceptance Rate**   $AR_{a'}$ represents the proportion of outputs $y$ sampled from $a'$ that satisfy the constraint $b(y)$. Formally $AR_{a'}$ is defined as the expected value of the constraint function $b(y)$ under the language model distribution $a'$, which is given by $AR_{a'} = \mathbb{E}_{y \sim a'}[b(y)]$. Higher $AR_{a'}$ indicates the sampler has a *faster inference speed*.

**Distributional Closeness**   In Theorem 2, we aim to minimize the KL divergence between the gold distribution $g$ and the distribution $g'$ produced by our framework. We can estimate the quality of GUARD, represented by $\text{KL}(g||g')$, by using the following identity:

$$\text{KL}(g||g') = \mathbb{E}_{y \sim g} \log \frac{g(y)}{g'(y)} = \mathbb{E}_{y \sim g} \log \frac{a(y)b(y)}{Z} \frac{Z'}{a'(y)b(y)} = \mathbb{E}_{y \sim g} \log \frac{a(y)}{a'(y)} - \log \frac{Z}{Z'}, \quad (10)$$

where the expectations relative to $g$ can be estimated by obtaining samples from $g$ through sampling from $a$ and filtering with $b$, and where the values of $Z$ and $Z'$ can be obtained by exploiting the equalities $Z = AR_a, Z' = AR_{a'}$ (see App. C.5) and by estimating these acceptance rates.

In practice, these computations were performed using the `disco` (Kruszewski et al., 2023) toolkit[18] and the estimations are made using $2^{20}$ (i.e., approximately 1M) samples from $a$. After filtering with $b$, we obtained around 2,300 samples from $g$ in the lexical constraints experiment ($AR_a = 0.0023$) and 5,000 samples from $g$ in the sentiment reversal experiment ($AR_a = 0.005$).

**Self-BLEU**   BLEU evaluates how similar a language model's output is to candidate references using $n$-gram matching. Similarly, Self-BLEU Zhu et al. (2018) measures the diversity of generated outputs by selecting $K$ samples from the model's output and calculating the average BLEU score of

---

[18]https://github.com/naver/disco

| Hyperparameter | Value |
|---|---|
| Base LLM $a(\cdot)$ | Gemma-2B[20] |
| Max sequence length | 30 |
| Learning rate | 1.41e-5 |
| Optimizer | Adam |
| DPG #samples per step | {2000, 5000, 10000} |
| SFT batch size per step | {16, 32, 64} |
| Sampling budget used with $a(\cdot|\text{CAP})$ | 10000 |
| Prompt used for warm-start proposal $a(\cdot|\text{CAP})$ | *Next sentence should contain "amazing".* |

Table 3: Hyperparameters used for the lexical constraint experiment.

each sample against the remaining $K - 1$ samples. In this paper, we measured Self-BLEU using a set of $K = 100$ outputs.

**Semantic Similarity**    First, we extract 500 outputs from each approach's guaranteed output distribution ($g$ or $g'$). We then randomly select 10,000 pairs of outputs from these 500 samples and embed them to measure their cosine similarity in the embedding space. The average of these similarity scores is then calculated. For the lexical constraint experiment, we encoded the Gemma-2B model outputs using the Gemma-7B model. In the sentiment reversal experiment, we encoded the GPT-2 model outputs using BERT.

## F    LEXICAL CONSTRAINT EXPERIMENT

**Additional details**    For the lexical constraint experiment, we use Gemma-2B (Riviere et al., 2024) to generate 30 tokens through ancestral sampling, starting with $\langle bos \rangle$ or a specific prompt in the case of the CAP baseline. We chose ancestral sampling because it generates from the LLM's original distribution without distortion, unlike top-$k$ or nucleus sampling. The sampling budget is defined as the number of samples drawn from proposal $a$, $a(\cdot|\text{CAP})$ or $a'$[19] for training purposes. For DPG, this is *#samples per step $\times$ step size*, while for SFT, it is *batch size $\times$ step size*. In the case of SFT, sampling is done only from $a$, so the expected training set size is *sampling budget $\times AR_a$*. For DPG, as the adaptive proposal gradually approaches $g$, the number of samples increases with each step during training. Naturally, $AR_{a'}$ is initially low, resulting in fewer samples, but this disadvantage can be mitigated by sampling from $a(\cdot|\text{CAP})$ in the early stage. Detailed hyperparameters are provided in Table 3, and the list of constraint-aware prompts used in the experiments can be found in Table 4.

**Additional analysis**    Based on the experiments discussed in the main body of the paper, we analyze how the phenomena observed in the analysis are reflected in the actual text. As shown in Table 5, texts generated from the DPG-based $g'$ feature the word "amazing" in various positions, discussing specific topics. However, when forced to include "amazing" through CAP, the generated texts tend to include the word "amazing" early on while remaining generic and without delving into specific subjects. This positional bias occurs not only with the string "amazing" but also with other strings such as "restaurant" and "world" as shown in Figure 9.

**Comparison to a heuristic sampler**    We performed a simple experiment using the heuristic sampler for keyword enforcement described in the *Enforcement example* paragraph of App. C.2, using "amazing" as the keyword to include in a 30-token generated text as in the experimental setting of Section 4.1. We define a sampler $s$ that checks whether we have already produced the string "amazing" within the first 29 tokens, and if not, forces the generation of "amazing" as the last token. With the Gemma-2B tokenizer, the string "amazing" can be generated with the single token [amazing] (along with [ama, zing], etc.), and we can easily design the sampler to produce this token as the 30th token if "amazing" does not appear earlier. For such a sampler $s$, we found that $\text{KL}(g||s) = 6.06$, which is a significantly larger divergence than $\text{KL}(g||g') = 0.633$ for warm-start DPG-based $g'$ as

---

[19]More precisely, the evolving $a'$ – in other words what is denoted by $\pi_\theta$ in Algorithm 2.
[20]https://huggingface.co/google/gemma-2b

| |
|---|
| Next sentence should contain 'amazing'. |
| Write sentences with the given words. 
 diagnosis: Assessment of microscopical and clinical parameters in the diagnosis of diabetes mellitus. 
 pandas: Column headings differ in spreadsheet that is merged with pandas data. 
 change: How to change the decimal separator in MS Word? 
 amazing: |
| Write a text containing "amazing", as shown below. |
| It is a really amazing day. |
| Amazing, amazing and amazing. |
| Truly amazing. |
| The following sentence contains the keyword 'amazing'. |
| This is an example of texts with the string 'amazing' |
| A paragraph with 'amazing': |
| Text examples with 'amazing'. |
| The following sentence contains the keyword 'amazing'. |

Table 4: Constraint-aware prompts used for the lexical constraint experiment.

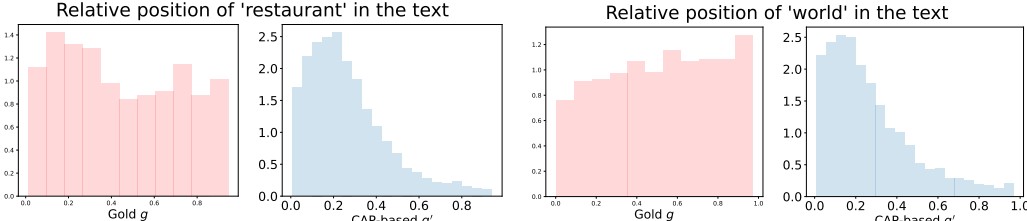

Figure 9: Additional positional bias analysis using the "restaurant" and "world" strings.

reported in Figure 4. More generally, the divergence $\text{KL}(g||s)$ is much greater than the divergences obtained for any $g'$ based on DPG or SFT, whose values are always smaller than 3. This result appears to be consistent with the behavior we had anticipated in App. C.2, namely the poor ability of $s$ to approximate $g$ due to the biased positioning of "amazing" in the generated sequence.

## G   SENTIMENT REVERSAL EXPERIMENT

**Additional details**   The lexical constraint experiment was set in an unconditional generation context (i.e., without any prefix), which makes the distortion due to prompting particularly severe. Therefore, it is necessary to thoroughly examine the phenomena that occur when $\text{KL}(g||g')$ increases in a conditional generation setting. For this experiment, we selected negative story openings from the ROCStories test set (Mostafazadeh et al., 2016). We retained only the first two sentences of all stories and chose five that had the lowest positive scores. The hyperparameters, constraint-aware prompts, and negative story openings used in this experiment are respectively shown in Tables 6, 7, and 8. To define $b(y)$, i.e., to accept or reject a generated story, we used a threshold criterion based on an existing emotion text classifier[22] using the "joy" class probability output. Specifically, we accepted stories where the "joy" class probability of the last sentence exceeded 0.98.

**Additional analysis**   The generated samples in Table 9 show patterns similar to those observed with the lexical constraint scenario. The stories obtained from the DPG-based $g'$ have a natural flow

---

[21]https://huggingface.co/msintaha/gpt2-finetuned-rocstories
[22]https://huggingface.co/michellejieli/emotion_text_classifier

| DPG-based $g'$ | CAP-based $g'$ |
|---|---|
| To be successful when tackling difficult problems and amazing your customers you need a space to think and solve these concerns. At CKWI, our expert and | 1. amazing ads on TV: They broke the advertising law and added more than 5 5 numbers to admit simulation! 2. Agata |
| Staten Island is world renowned for its dog-friendly beaches and amazing shops. The zoo park is always a hit for residents and tourists alike. Visits | 3 most amazing facts about marine animals. paragraphs: paragraph, paragraph. extension: zoom calculations: result of calculation application programming: application |
| Welcome Guest,
We do believe having a great relationship with clients is invaluable, therefore we strive to make your playing experience as comfortable, amazing and exciting | Things wouldn't be so amazing without all that PERFECTNESS which we call 'Love'......
Thanks Vernachael, I'm dropping by |
| Join us on an amazing journey In Southern Uzbekistan.
17 days Arrive in Tashkent and upon arrival, we will pick you up from the Airport | I agree, that is truly amazing. I like the fact it is all customisable.
That's nice bet it stinks out the exhaust fumes |

Table 5: Example of generated samples for the lexical constraint experiment. Red-colored text indicates the occurrence of the required string "amazing".

| Hyperparameter | Value |
|---|---|
| Base LLM $a(\cdot)$ | GPT2-Finetuned-ROCStories[21] |
| Max sequence length | 80 |
| Learning rate | 1.41e-5 |
| Optimizer | Adam |
| DPG #samples per step | 10000 |
| SFT batch size per step | 64 |
| Sampling budget used with $a(\cdot|\text{CAP})$ | 10000 |
| Prompt used for warm-start proposal $a(\cdot|\text{CAP})$ | *This is a happy ending story:* |

Table 6: Hyperparameters used for the sentiment reversal experiment.

like the original $g$, while the samples obtained from CAP exhibit abrupt sentiment changes, losing specificity in the process. This leads to significant differences in KL divergence between $g$ and $g'$.

**A note about the Story Infilling task of (Hu et al., 2024)**  Our sentiment reversal experiment bears some connections with the Story Infilling task from Hu et al. (2024, Section 4.2), where the authors consider the task of generating the middle (fourth) sentence of a story, given the first three and the last (fifth) sentence of the story, using the ROCStories dataset for training and for test. They use the GFlowNet amortization framework (Bengio et al., 2021) to fine-tune their base model towards generating the middle sentence. Our task is similar in spirit, because we try to complete the start of a story in such a way that the story ends on a positive sentence (instead of ending with a fully specified sentence). In other words, our version of the task can be considered as *soft* Story Infilling. Additionally, the base model $a$ we use is GPT-2 fine-tuned on the ROCStories training set, as in Hu et al. (2024). However, our approach is quite different both in the goal and in the method used. We are trying to obtain a sampler $g'$ that produces texts coming from a distribution close to the gold distribution $g$ — where $g$ is the distribution $a$ filtered by the positive-ending constraint — and we insist on only sampling texts with a positive ending. In contrast, they do not try to filter texts from $a$ by the constraint that they end on a specific sentence, but only use $a$ as the starting point for their GFlowNet amortization, trained on the ROCStories dataset. They evaluate their results in terms of the similarity of the generated middle sentence to the middle sentence in the ROCStories test set, while we evaluate our results in terms of the divergence with the gold distribution $g$ and of the efficiency of the sampler in producing positive endings.

| |
|---|
| This is a happy ending story: |
| In the end, it was a story that had a truly happy ending, |
| This is my positive ending story, |
| Eventually, it turned into a really fantastic day: |
| Here is an example of a paragraph starting up with a negative sentiment but ending positively: |
| Happy ending story: |
| Eventually, it turned into a really happy day! |
| There was a little bit of bad luck, but it was a happy day! |
| I was so lucky even there were some obstacles. |

Table 7: Constraint-aware prompts used for the sentiment reversal experiment.

| |
|---|
| My uncle couldn't afford health care. He got sick last year. |
| Kevin's dad had raised him alone after his mom left them. When his dad grew old and ill, Kevin was unhappy. |
| Allie came home late last night. Her mother was very mad at her. |
| Mike was a bad kid. He never followed his parents advice. |
| Bobby registered his son for tee ball. His son wasn't having a good time. |

Table 8: Negative story openings used for the sentiment reversal experiment.

# H  DISCUSSION OF ADDITIONAL SAMPLING TECHNIQUES: IMH AND QRS

This section contrasts the efficiency of GUARD at inference time with one MCMC technique, IMH (Robert & Casella, 2004), and one variant of rejection sampling, QRS (Eikema et al., 2022). It compares outputs from these different sampling methods, starting with a specific proposal distribution $a'$: rejection sampler $g'$ as in GUARD, IMH sampler, and QRS sampler. QRS is an extension of rejection sampling that offers two benefits over IMH as a sampler for GUARD: (1) It produces outputs $y$ that more closely approximate the target distribution $g$; (2) It still enables tractable measurement of the distance from $g$, similarly to rejection sampling in Algorithm 1.[23] by allowing quantification of the output probability of $y$.

QRS implements the acceptance criterion corresponding to accepting $y$ with probability $\min(1, \frac{g(x)}{\beta a'(x)})$, where $\beta$ is a positive hyperparameter with the following properties:

1. As $\beta$ approaches 0, QRS becomes equivalent to the GUARD form of rejection sampling (Algorithm 1).
2. As $\beta$ increases, QRS more closely approximates $g$, but at the cost of additional inference time.

This comparison permits an analysis of the trade-off between sampling accuracy and computational cost across these different methods.

## H.1  EXPERIMENTAL SETTING

Directly comparing the GUARD sampler with MCMC methods like IMH is challenging. When we run the IMH algorithm for a certain number of steps $n$, we obtain a distribution denoted as $p^{(n)}$. While we can sample from $p^{(n)}$, we cannot evaluate $p^{(n)}(x)$ for a given $x$ (see Eikema et al. (2022)), which presents a significant obstacle to estimating the KL divergence.

---

[23]Note that, in the experiments below, in order to compare IMH and QRS, we use an approximate measure available for both.

| DPG-based $g'$ | CAP-based $g'$ |
|---|---|
| My uncle couldn't afford health care. He got sick last year. His aunt bought him medicine for his sick days. My aunt bought him some and started treating him. He has much better health than he had before. He is very grateful, and is doing better everyday. | My uncle couldn't afford health care. He got sick last year. I hope someday he gets better. I hope my grandma gets better soon. Best of luck! Ever! |
| My uncle couldn't afford health care. He got sick last year. The only way around it was by prescription. A doctor arranged to have me take antiretrovirals and a lot of painkillers. He had a wonderful 18 month treatment and I feel much better today. | My uncle couldn't afford health care. He got sick last year. Luckily he got home healthy. It was wonderful to be able to share our kitchen with our children. It feels so much better than we ever knew. |
| Mike was a bad kid. He never followed his parents advice. He was misbehaving and frightened. He tried to learn from his own mistakes. Eventually, he was a very good boy. He is very grateful. This gift has made a huge difference to my son. | Mike was a bad kid. He never followed his parents advice. However, he did learn a new trick. He learned how to play the guitar. I can't wait to see how he plays! I hope all that gets passed on to Mike someday! |
| Allie came home late last night. Her mother was very mad at her. She put a basket of cookies on the kitchen table. She went to the kitchen to make her dinner. She made several small cuts, then her mother ate her favorite slice. So she and her mom were very happy! | Allie came home late last night. Her mother was very mad at her. She decided to surprise her grandma and surprise her. Allie packed her things and got ready to go. Everyone had a great day! |

Table 9: Example of generated stories for the sentiment reversal experiment. Blue-colored text corresponds to the imposed negative opening prefix, and red-colored text indicates the final sentence of the story, on which sentiment positivity is required.

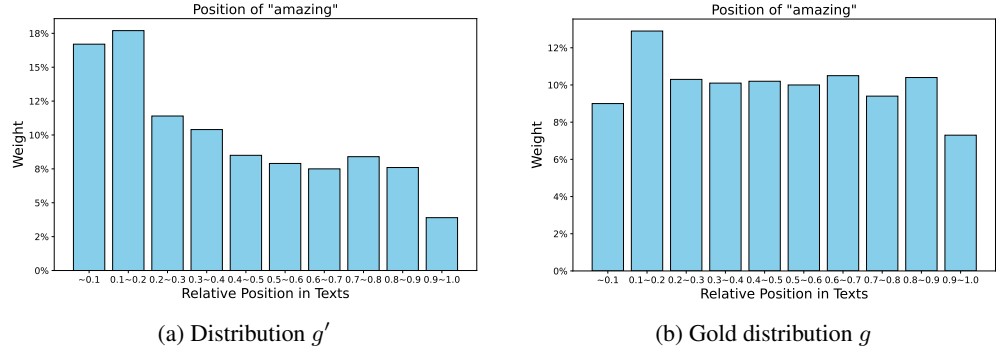

(a) Distribution $g'$          (b) Gold distribution $g$

Figure 10: Two base (projected) distributions for comparing with QRS and IMH. Starting from the distribution in (a), the goal of each statistical sampler is to generate a similar distribution to (b). For a clearer analysis of the phenomenon, we conducted experiments using a proposal $a'$ with a relatively large divergence, resulting in $\text{KL}(g||g') \approx 2$.

Although we cannot directly calculate $p^{(n)}$, we could estimate it using the law of large numbers. However, this becomes problematic when the sample space $X$ is very large, as in text generation cases. In contrast, if our sample space $X$ was small (e.g., $X = \{1, 2, ..., 10\}$) and we had 10,000 samples from a given sampler $\pi$, we could estimate $\pi(2)$ by counting the relative frequency of 2's in the sample, and similarly for any $x$ in this space.

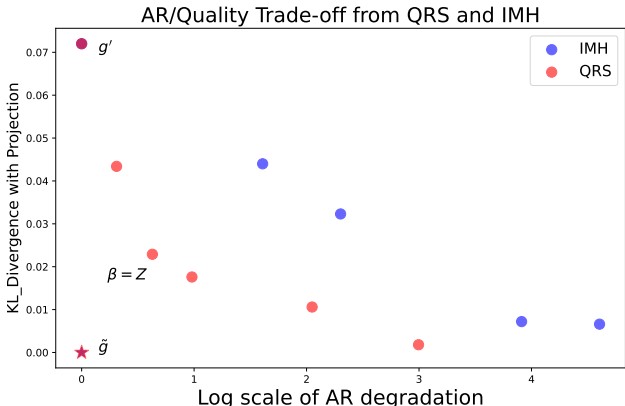

Figure 11: Starting from the top-left point (i.e., initial $g'$), this figure illustrates the improvement of the projected KL over $\mathrm{KL}(\bar{g}||\bar{g}')$ for both IMH and QRS and the associated additional inference costs. We note that QRS achieves a reasonable compromise when $\beta = Z$.[25]

Based on this insight, we propose the following approach for providing at least an approximate estimation of $\mathrm{KL}(p||p^{(n)})$:[24]

1. We build a small finite partition $\bar{X}$ of 10 bins for the textual space $X$, with a projection mapping $f(x) = \bar{x}$ from the large space $X$ to the low-dimensional space $\bar{X}$.

2. We generate a sample of 1000 points from $p$ and use $f$ to project these points into points $\bar{x}$ of $\bar{X}$. We get an estimate $\bar{p}(\bar{x})$ for any $\bar{x}$ in $\bar{X}$.

3. We do the same for $p^{(n)}$, obtaining estimates $\bar{p}^{(n)}(\bar{x})$.

4. We can then estimate $\mathrm{KL}(\bar{p}||\bar{p}^{(n)})$.

5. Then, by considering different values of $n$ (but keeping the projection $f$ fixed), we can monitor how $\mathrm{KL}(\bar{p}||\bar{p}^{(n)})$ evolves with $n$. It is expected to decrease with $n$.

In this experiment, we focus on the lexical constraint scenario. Then, for the projection $f$, we can use the position of "amazing" in the generated text. We can project the texts from $g$ and $g'$ based on which of the 10 bins the relative position of "amazing" falls into (see Figure 10). Using this method enables us to compute a form of KL divergence for IMH and to use this form for comparison with QRS as shown in Figure 11.

## H.2 EXPERIMENTAL RESULTS

Figure 11 illustrates the quality/AR trade-off results. Both QRS and IMH demonstrate improved output quality at the cost of additional inference time. However, QRS consistently Pareto-dominates IMH across the tested hyperparameters.

In the case of QRS and especially IMH, the quality gains come at a high cost in terms of additional inference time: to halve the projected KL divergence — which is a lower bound of the true KL divergence, see Footnote 24 — QRS requires 1.36 times the inference cost of Algorithm 1, while IMH requires 10 times the cost. Moreover, to reduce the KL divergence to 10% of the original KL, QRS uses almost 10 times the inference cost of Algorithm 1, while IMH consumes 50 times this cost. This experiment demonstrates that the random walk nature of MCMC methods can struggle to efficiently mimic the gold distribution. This result highlights why accurately approximating $g$ with the proposal distribution is crucial: it can lead to more efficient sampling that diverges less from the target distribution without incurring the extreme computational costs associated with methods like IMH.

---

[24]Actually, this technique provides a lower-bound of the true KL divergence, a consequence of the Data Processing Inequality (DPI) of Information Theory (Polyanskiy & Wu, 2023).

[25]IMH performs a random walk $n$ times on an initially accepted sample to improve its quality, where $n$ is a hyperparameter, and where the last $y$ in the random walk is returned. For comparison with the rejection-based samplers, we therefore consider the "acceptance rate" of IMH to be $1/n$.

