# OpenReview forum: "Guaranteed Generation from Large Language Models"
_ICLR.cc/2025/Conference — ICLR 2025 Poster_

### Official Review · Reviewer_vMCq · 2024-10-23

**Soundness:** 4
**Presentation:** 4
**Contribution:** 4
**Rating:** 8
**Confidence:** 3

**Summary:**

This work proposes the GUARD framework, which guarantees strict constraint satisfaction for generated outputs of LLMs.
It utilizes rejection sampling at inference to guarantee constraint satisfaction as well as a variant of DPG to tune the language model towards a policy close to the "gold standard" policy to ensure distributional closeness during training.

**Strengths:**

I found the problem and the proposed solution very interesting and intuitive.
The experiments seem reasonable, the results good and I see no major evaluation missing.
The auxiliary investigations in Fig 4 & 7 as well as Tab. 1 and 2 are very insightful.

**Weaknesses:**

I strongly agree that the major limitation of this approach in obtaining a filter b (also called verifyer in other areas), which I think is one of the most important research areas around LLMs right now.
However, I agree with the last paragraph in the main text, that this is out of scope for this work and any improvements in that direction will directly improve GUARD.

Minor:
* I understand why the baseline method described in line 172-176 and in App. A.2 has severe drawbacks, but for the simple constraint in the first experiment in Sec. 4.1, it would actually be a suitable baseline to compare to.
Also I could see that it performs better than DPG with CAP in this setting, as the distribution is less degenerate I guess.
* It would be nice to have the two experiments performed with more than one model each, but I understand the computational demand arising from this.
* The claim in the third contribution is a bit too strong for me. GUARD itself does not improve the acceptance rate, it depends on the way a' is chosen right? So the improved DPG leads to higher acceptance rates.

Remarks (just as info for camera ready):
* SFT is not introduced on first occurance in line 200, but in 255/256.
* The order of Fig. 3 and 4 is reversed

**Questions:**

* What is V^* in line 119? It is never defined and I can only find it used once in line 801 where I don't get its usage.
* What is "ancestral sampling" - line 308?

---

> ### Author Response · Authors · 2024-11-20
> **Response to Reviewer vMCq**
>
> Thank you for your review and constructive feedback on our work! We appreciate your recognition of the strengths and value of our paper. We address below the points raised in your review.
>
> **Comparison with the heuristic sampler from Appendix A.2**
>
> We assume this question is referring to the heuristic algorithm presented in the *Enforcement example* paragraph in Appendix A.2 (if not, please correct us). While this algorithm has clear expected drawbacks, we agree it would be interesting to verify its performance numerically in the lexical experiment to include “amazing”. We tested this, in the bounded-length setting of Section 4.1 with a 30-token generation length. In the heuristic sampler, we check whether we have already produced the string "amazing" within the first 29 tokens, and if not, force the generation of "amazing" as the last token. With the Gemma-2B tokenizer, the string "amazing" can be generated with the single token [amazing] (along with [ama, zing], etc.) and we can easily design the sampler to produce this token as the 30th token if “amazing” does not appear earlier. For such a sampler $s$, we found that $KL(g||s)=6.06$, which is a significantly larger divergence than $KL(g||g')=0.633$ for the $a'$ based on warm-start DPG.
>
> **GUARD itself does not increases the acceptance rate, but $a’$ does**
>
> We agree that our formulation in the third contribution could lead to some confusion. We will modify it to clarify that it is indeed the choice of $a’$ that leads to improving the acceptance rate, rather than the GUARD framework itself.
>
> **Remarks for the camera-ready version**
>
> Thank you for pointing out these mistakes, we will correct them in the updated version of the submission.
>
> **Question 1 — What is** $V^*$ **in line 119?**
>
> This is the “Kleene star” notation for the set of all finite sequences based on the token vocabulary $V$.
>
> **Question 2 — What is "ancestral sampling"?**
>
> The term “ancestral sampling” denotes basic sampling from the model — namely sampling with temperature 1, directly reflecting the probability distribution associated with the model without any distortions or truncation (unlike techniques such as top-$k$ or nucleus sampling).

---

> > ### Comment · Reviewer_vMCq · 2024-11-21
> >
> > Q2: Ah ok, I usually see the term "multinomial sampling" used for that, maybe check in related work what is more widely used.
> >
> > Thank you for the rebuttal, I will keep my rating

---

> > > ### Author Response · Authors · 2024-11-22
> > > **Response to Reviewer vMCq**
> > >
> > > Thank you for reading our rebuttal and getting back to us. Both ancestral sampling and multinomial sampling seem to be used -- for example, both are mentioned on Huggingface's generation strategies page: https://huggingface.co/docs/transformers/generation_strategies#multinomial-sampling. As ancestral sampling (a term that originates in probabilistic graphical models) emphasizes more the autoregressive nature of the process, we prefer to keep it.

---

### Official Review · Reviewer_hVqd · 2024-10-25

**Soundness:** 3
**Presentation:** 2
**Contribution:** 2
**Rating:** 5
**Confidence:** 4

**Summary:**

This work proposes GUARD, a method that combines an autoregressive proposal distribution with rejection sampling to guarantee the satisfaction of hard constraints in generated text while keeping the output close to the language model’s original distribution.

**Strengths:**

- **Clear Motivation and Intuition:** The authors provide clear intuitions about the need for distribution-preserving constraint satisfaction in language models and the challenges this entails. They thoroughly motivate their approach for achieving strict control over generations without substantially deviating from the original model’s output distribution.
- **Theoretical Rigor:** The proposed method is supported by theoretical principles, providing a mathematically grounded mechanism that guarantees strict constraint satisfaction while preserving the original model’s output distribution.
- **Efficiency in Performance:** Empirically, the method achieves improved inference efficiency by increasing the acceptance rate compared to the original model while maintaining minimal distributional distortion.

**Weaknesses:**

**Theory:** Although well-motivated, the novelty of the approach is somewhat unclear, as it largely combines established alignment techniques with a naïve rejection sampling method to achieve guaranteed generation from language models:
- First, the authors propose a specific prompt (CAP) and/or a fine-tuned model (SFT, DPG), denoted as $a’$, to approximate the gold distribution $g$. These methods are commonly used to align model outputs with desired behaviors. However, there is no guarantee that each method minimizes $KL(g || a’)$, even though the authors stress its importance for both quality and efficiency (lines 249-250).
- Second, $a’$ is used to generate answers that are then filtered through rejection sampling, which guarantees that constraint $b$ are satisfied but which has already been discussed in prior work (e.g., Yang and Klein, 2021).

**Presentation**: In several parts, the work lacks precision and conciseness:
- First, critical aspects such as the approximation and minimization of $KL(g || a’)$ remain unclear (see question 1 below).
- Second, essential insights into the efficiency of the method and the quality of the generated answers are insufficiently addressed (see questions 2 and 3 below).
- Third, the method assumes that constraints $b$ can be designed such that it respects the desired behavior. Since this is a core assumption of this work, it should not be considered “out of scope”.
- Fourth, consistency in terminology could be improved. For example, while “autoregressive model” is frequently referenced, the abbreviation “ARM” is introduced only in Section 2.2 and used inconsistently thereafter.

---
K. Yang and D. Klein. 2021. Fudge: Controlled text generation with future discriminators. arXiv preprint arXiv:2104.05218.

**Questions:**

- How is $\mathrm{KL}(g || a') =\mathrm{KL}(g || g') - \mathrm{AR}_{a'}$ computed? The gold distribution $g$ is not accessible and the acceptance rate $\mathrm{AR}$ is infeasible to compute as it requires considering all possible output sequences, which is of complexity $\mathcal{O} (| \mathcal{V} |^{T})$.

Also, since $\mathrm{KL}(g || a') > 0$, questions remain about the quality and efficiency of the proposed methods, especially in comparison to other baseline techniques:
- How do generated answers compare to baselines? Does $g'$ still capture the same capabilities as the original model $a$?
- How efficient is the approach compared to baselines? How frequent are answers from $a'$ accepted compared to the answers from $a$?

---

> ### Author Response · Authors · 2024-11-20
> **Response to Reviewer hVqd (part 1)**
>
> Thank you for your detailed feedback. We address below each of the points raised in your review.
>
> **Minimization of $KL(g || a')$**
>
> As you rightfully pointed out, our goal is to minimize $KL(g || a')$ to obtain a 'good' $a'$ (as justified by Theorem 2 which links this quantity to both efficiency and closeness to $g$). In fact, both SFT and DPG seek to minimize $KL(g || a')$ by definition, as detailed below.
>
> The SFT loss corresponds to the cross-entropy of the model $a'$ to be learned, using samples from $g$, i.e., $-\mathbb{E}\_{y \sim g} \log a'(y)$. Minimizing this term with respect to $a’$ is equivalent to minimizing $KL(g || a')$ since $KL(g || a') = \mathbb{E}\_{y \sim g} \log \frac{g(y)}{a'(y)} = \mathbb{E}\_{y \sim g} \log g(y) -\mathbb{E}\_{y \sim g} \log a'(y) = H_g - \mathbb{E}\_{y \sim g} \log a'(y)$ where $H_g$ is the entropy of $g$ which is a constant independent of $a’$.
>
> DPG’s objective is also to minimize the KL divergence (or equivalently, the cross-entropy) between the target distribution $g$ and the learned policy $a' = \pi_{\theta}$ as mentioned in Section 3, paragraph *Approximating the target distribution*: “This method samples $y$ from a proposal $a'$ initialized with $a$ and updates $a'$ by performing gradient descent on $KL(g||a')$.” This property is further detailed in Section 3.2 of Parshakova et al (2019). The derivation can be summarized as follows: $\nabla_{\theta} KL(g||\pi_{\theta}) = \nabla_{\theta} (H_g - \mathbb{E}\_{y \sim g} \log \pi_{\theta}(y)) = -\mathbb{E}\_{y \sim g}\nabla_{\theta} \log \pi_{\theta}(y) = -\mathbb{E}\_{y \sim \pi_{\theta}} \frac{g(y)}{\pi_{\theta}(y)} \nabla_{\theta} \log \pi_{\theta}(y)$, where the last step is obtained by applying importance sampling using $\pi_{\theta}$ as proposal. Performing gradient descent on $-\mathbb{E}\_{y \sim \pi_{\theta}} \frac{g(y)}{\pi_{\theta}(y)} \nabla_{\theta} \log \pi_{\theta}(y)$ leads to the DPG update rule shown at Line 21 of Algorithm 2.
>
> We will update the paper to clarify the $KL(g || a')$ minimization property in SFT and DPG.
>
> **Rejection sampling in FUDGE**
>
> While rejection sampling is indeed briefly discussed in Yang and Klein (2021) as a possible way to improve FUDGE, it is not explored in that paper and only mentioned “in passing” as a potential extension. In contrast, we provide in our paper an in-depth theoretical and empirical investigation of the use of rejection sampling for guaranteed generation.
>
> **Design of $b$**
>
> We agree that the choice of $b$ is important (as pointed out in our *Limitations* paragraph in the *Conclusion* section). However, we still believe that the investigation on the design of $b$ is orthogonal to the focus of our paper — namely, the GUARD approach — as it implies a modification of the core target $g$. In other words, modifying $b$ would affect the target distribution $g$, while our paper's primary objective is to find a guaranteed sampler that approximates $g$ for a given $b$. Studying the choice of $b$ may then warrant a paper of its own, and any improvement of $b$ will directly translate into improving GUARD, as reviewer vMCq also noted.

---

> ### Author Response · Authors · 2024-11-20
> **Response to Reviewer hVqd (part 2)**
>
> **Consistency in ARM abbreviation usage**
>
> Thank you for pointing this out, we will correct this.
>
> **Computation of $KL(g || a')$, $KL(g || g')$ and $AR_{a'}$**
>
> According to Theorem 1, it is indeed typically impossible to find an autoregressive model that matches distribution $g$ exactly, but we can still obtain gold samples from $g$ by first sampling from $a$ and then filtering with $b$ (though this may be highly inefficient in case of low $AR_a$). These samples are used to estimate $KL(g || g')$ as detailed in Equation 10 in Appendix C. The estimate of $KL(g || a')$ is obtained in the same way. Similarly, $AR_{a'}$ which is equal to $\mathbb{E}\_{y \sim a'} [b(y)]$ can be estimated by drawing samples from $a’$. The consistency of these estimates can also be verified using Theorem 2 which links these three quantities.
>
> **Quality of generated answers and capabilities of $g'$ wrt $a$**
>
> There are two types of deviations of $g'$ relative to $a$. The first deviation is unavoidable and corresponds to an *intrinsic distortion*, which is the fact that $g$ itself has to deviate from $a$ to some extent to enforce the constraint $b$ (although this deviation is minimal as explained in Appendix A.1). The other deviation is related to the *approximation error*, namely the value $KL(g || g')$ that we are trying to minimize.
>
> Note that in some cases the constraint $b$ makes it impossible to maintain the capabilities of the original model (e.g., if $b$ corresponds to a strict notion of harmlessness, it may not preserve the ability of the model to be helpful in certain limit cases), while in some other cases (e.g., if $b$ enforces a polite tone) the capabilities of $a$ are mostly preserved in $g$ (and, in turn, $g'$). Because of these variations, that depend on the nature of $b$, we concentrate on the approximation error $KL(g || g')$ rather than on the intrinsic distortion $KL(g || a)$ in our experiments.
>
> We also provide examples of generated text with DPG-based $g'$ and CAP-based $g'$ in Tables 5 and 9 in the Appendix.
>
> **Efficiency of the different approaches**
>
> The efficiency at inference time is measured by the acceptance rate of $a’$. The acceptance rate improvement of GUARD over applying rejection sampling on $a$ is up to 180x for the lexical constraint experiment and 60x for the sentiment reversal experiment, as detailed in Sections 4.1 and 4.2, respectively. In more intuitive terms, using $a$ for rejection sampling leads to accepting 1 sample out of 435 drawn samples for lexical constraints and 1 sample out of 200 drawn samples for the sentiment reversal. In comparison, using GUARD, this goes up to accepting approximately 1 sample out of 2 drawn samples and 1 sample out of 3 drawn samples, respectively. We will update the paper to include these numbers, which we agree are easier to interpret.

---

> > ### Author Response · Authors · 2024-11-24
> > **Follow up to our response to Reviewer hVqd**
> >
> > Dear Reviewer hVqd,
> >
> > As the end of the discussion period is nearing, we would like to follow up to see if our detailed response addressed your concerns. We would also be happy to answer any further questions you have. Thank you again for your time and helpful feedback.

---

> ### Comment · Reviewer_hVqd · 2024-11-26
>
> Thank you for the detailed response!
>
> - I appreciate the clarification on rejection sampling and the fact that SFT and DPG minimize your objective. Could you also explain how CAP aligns with this?
> - I still do not fully agree with the statement that investigating the design of $b$ is orthogonal work. If it is impossible to construct a meaningful $b$ for real-world applications, finding a guaranteed sampler that approximates $g$ for a given $b$ adds no practical value.
> - Regarding the quality of the answers, I am mainly concerned that $g'$ might underperform on general benchmarks compared to $a$ (e.g., solving math problems when constrained to avoid generating political statements). This concern arises because fine-tuning alters the original model, and the CAP prompts themselves appear to be quite ambiguous (e..g. generating a sentence that contains *"amazing"* involves a prompt like *"Write sentences with the given words. diagnosis: Assessment of microscopical and clinical parameters in the diagnosis of diabetes mellitus. pandas: Column headings differ in spreadsheet that is merged with pandas data. change: How to change the decimal separator in MS Word? amazing:"*)
> - Regarding efficiency, does this imply that 435 samples are required to obtain one additional training sample for SFT and DPG? How many samples did you generate in total for training? Similarly, for computing  $KL(\cdot || \cdot)$, how many samples were generated to approximate the divergences between $g$, $g'$ and $a'$?

---

> > ### Author Response · Authors · 2024-11-27
> > **Response to Reviewer hVqd (part 1)**
> >
> > Thank you for your additional questions, which we address below.
> >
> > **Q1. While SFT and DPG aim to minimize** $KL(g||a’)$**, how does CAP align with this?**
> >
> > Prompting, as used in CAP, provides an inexpensive way (without parameter tuning) for obtaining an ARM $a'$ that tends to satisfy the constraint much more often than the original model $a$ does. Such an $a'$ is closer to $g$ than $a$ is, in terms of the acceptance rate with respect to $b$ — in formal terms $\mathbb{E}_{y\sim a'} b(y)$ is close to $1=\mathbb{E}\_{y\sim g} b(y)$, while $\mathbb{E}\_{y\sim a} b(y)$ is typically much smaller. Despite this, as our experiments show (see Figures 4 and 7 in the revised version of our submission), CAP tends to have a worse $KL(g||a')$ than those $a'$ that are obtained by either SFT or DPG — an interesting observation in its own right, we think. This implies that when CAP is used, the constraint is enforced by strongly biasing the generation, i.e., at the cost of a high $KL(g||a')$.
> >
> > However, when the initial acceptance rate is low and training DPG from $a$ is slow in the early stages (due to the scarce gradient updates), CAP does help. Although the CAP samples are biased, it is very advantageous to obtain $y$’s that satisfy the constraint, and this yields a “warm” initial policy $a'$ with a high acceptance rate $AR_{a'}$ to start DPG (this is described lines 3-10 in Algorithm 2 of the appendix). In summary, although the CAP-based $a'$ has some bias, its high acceptance rate allows us to obtain rich signals about $g$ during the training of DPG, which explains the good performance of GUARD with warm-start DPG.
> >
> > **Q2. Should the design of $b$ be considered as out of scope for this paper?**
> >
> > While we understand your concern with respect to the importance of $b$, we do believe in the value of decoupling the design of $b$ (i.e., what you suggest) from the implementation of the related target distribution $g$ through a generator (i.e., what we do in the paper). While the latter is a mostly understudied problem in the literature (see our updated *Related Work* section in the revised PDF), the former has been studied extensively through numerous works on classifiers, reward models or verifiers. For example, there exists strong $b$’s for sentiment-related tasks or for safety constraints (see for example the recent OpenAI moderator [1], LlamaGuard [2], or ShieldGemma [3]). More generally, for a custom task, one could obtain $b$ through a supervised classifier trained from annotated data or through a zero-shot classifier [4]. Given the large body of literature on the topic, we believe that assuming the ability to find a $b$ for real-world applications is not overly unreasonable.
> >
> > [1] https://openai.com/index/using-gpt-4-for-content-moderation/
> >
> > [2] https://huggingface.co/meta-llama/Llama-Guard-3-8B
> >
> > [3] https://ai.google.dev/gemma/docs/shieldgemma
> >
> > [4] Zhiqiang Wang, Yiran Pang, Yanbin Lin: Large Language Models Are Zero-Shot Text Classifiers. arXiv:2312.01044 (2023)

---

> ### Author Response · Authors · 2024-11-27
> **Response to Reviewer hVqd (part 2)**
>
> **Q3. Does the fine-tuning or prompting of** $g'$ **lead to a loss of general performance in comparison to $a$?**
>
> First, we agree that using CAP with the few-shot prompt mentioned in your question would lead to a loss of general performance in the resulting $g'$. This is one of the main downsides of prompting, which tends to bias and restrict the scope of the generated output, as pointed out in our answer to Q1.
>
> For a $g'$ obtained through fine-tuning, using DPG or SFT, the loss of general performance is related to the *intrinsic distortion* and to the *approximation error* that we discussed in our previous response. Depending on the constraint $b$ of interest, certain capacities from $a$ will not be present anymore in $g$ — which relates to the *intrinsic distortion*. To follow the example given in your question, using a $b$ that forbids the mention of some political parties *X*, *Y*, and *Z* would make it impossible for $g$ to solve a math problem such as “In a small town, there are three political parties, the *X*, the *Y* and the *Z*, each party has a certain number of supporters, …”. This would then naturally be reflected in the performance of the DPG/SFT-based $g'$ on this specific math problem.
>
> However, on math problems which do not mention parties *X*, *Y*, and *Z*, the performance of $g$ would be equivalent to that of $a$. Then, if the *approximation error* between $g$ and $g'$ is low (as it is the case for DPG/SFT-based $g'$ in the lexical constraints and sentiment reversal settings), the performance of $g$ (and thus $a$) will be preserved in $g'$.
>
> **Q4. Details about the number of samples used for training and evaluation.**
>
> You are right that obtaining one additional sample for SFT does require 435 samples from $a$, which is also our argument against using SFT due to its inefficient nature. However, for DPG, this is only true at the *beginning* of the training of DPG, (when $\pi_\theta = a$ in Algorithm 2, implying few effective gradient updates on line 21), because later on, $\pi_\theta$ gets closer to $g$ and therefore produces more actual updates — this is what makes DPG an *adaptive* algorithm. When warm-starting DPG with CAP, the samples from $\pi_\theta$ are actually useful from the start, as we mentioned in the answer to your first question. The number of samples used in training (sampling from $a$) is defined as the "sampling budget" in Figures 3 and 5 from the revised PDF, and we experimented with up to 800,000 and 1,500,000 samples, respectively.
>
> For the evaluation, we sampled $2^{20}$ (i.e., approximately 1,000,000) new samples from $a$ to perform the KL estimation — while this is costly, we typically only perform these estimations for scientific analysis purposes. Then, in the lexical constraints scenario (where $AR_a$ = 0.0023), this leads to approximately 2,300 samples from $g$. For the sentiment reversal experiment (where $AR_a$ = 0.005), this yields 5,000 samples from $g$. We have added these details in Appendix D in the revised submission.
>
> We hope that our responses helped clarify the concerns you raised, and we thank you again for your valuable engagement in this exchange.

---

> > ### Author Response · Authors · 2024-11-29
> > **Gentle reminder before the end of the discussion period**
> >
> > Dear Reviewer hVqd,
> >
> > Thank you once again for taking the time to react to our initial responses. We are aware that the timing since our follow-up responses is tight but as the end of the discussion period is approaching, we would like to know if we have addressed the concerns you raised in your latest questions. If you feel that it is the case, we would be very grateful for this to be reflected in score adjustments to acknowledge our responses.

---

> > > ### Comment · Reviewer_hVqd · 2024-12-01
> > >
> > > Thank you for the response.
> > >
> > > **Q1**
> > > > Prompting, as used in CAP, provides an inexpensive way (without parameter tuning) for obtaining an ARM $a'$ that tends to satisfy the constraint much more often than the original model
> > >
> > > To me, it remains unclear how to obtain such a prompt. Moreover, this does not provide an argument for guaranteeing that $KL(g || a’)$ is minimized.
> > >
> > > **Q2**
> > >
> > > Your arguments are convincing. They should be discussed in more detail in the main paper.
> > >
> > > **Q3**
> > > > the performance of $g$ would be equivalent to that of $a$
> > >
> > > Claiming such statements without supporting evidence is not convincing. It is well known that fine-tuning and instruction-prompting can significantly impact a model’s performance.
> > >
> > > **Q4**
> > >
> > > While I appreciate the insights, the need to generate such a large number of samples for a relatively simple constraint amplifies my concerns about the practicality and scalability of the approach.
> > >
> > > Given this remaining concerns, I maintain my initial rating.

---

> ### Author Response · Authors · 2024-12-02
>
> Thank you once again for taking the time to follow up on our previous responses.
>
> **Q1. To me, it remains unclear how to obtain such a prompt. Moreover, this does not provide an argument for guaranteeing that** $KL(g||a')$ **is minimized.**
>
> To clarify any misunderstanding, we did *not* say that the $a'$ resulting from prompting guarantees a low $KL(g||a')$, but only that it tends to *satisfy the constraint* much more often than the original model (which is very different) without any training cost, namely that $AR_{a'}$ tends to be better than $AR_{a}$. In fact, we explicitly said the opposite: “CAP tends to have a worse $KL(g||a')$ than those $a'$ that are obtained by either SFT or DPG […]. This implies that when CAP is used, the constraint is enforced by strongly biasing the generation, ie, at the cost of a high $KL(g||a')$”. Experimentally, looking at Fig. 4, we do see that every CAP-based $a'$ have $KL(g||a')$ larger than 10, even worse than $KL(g||a)$ which is around 6. Please note that in order to read $KL(g||a')$ from this figure, one needs to look at the intersection between the dotted line and the x-axis. For instance, the leftmost pale-blue CAP point lies on the dotted line that intersects this axis for $x \approx 10$. By comparison, all the fine-tuned versions of $a'$, either through SFT or through one of the two DPG variants, has a $KL(g||a')$ lower than 4. The same trend can be observed in Fig. 7, for the sentiment-reversal experiment.
>
> On the other hand, what we indicated in our previous response is that a CAP-based $a'$ is a good *starting point* for the warm-start version of DPG (namely DPG initialized with $a(\cdot|CAP)$). Warm-start DPG yielded better results for $KL(g||a')$ than DPG both in Fig. 4 and more so in Fig. 7, and with a quicker convergence than DPG (see Fig. 3). In both experiments, the best $KL(g||a')$ result is obtained for warm-start DPG, at around 1.4 in Fig. 4 and 1.9 in Fig. 7.
>
> **Q2. Your arguments are convincing. They should be discussed in more detail in the main paper.**
>
> Thank you! We will update our paper with these arguments as soon as we are allowed again to update the paper.
>
> **Q3. Claiming such statements (”the performance of** $g$ **would be equivalent to that of** $a$**”) without supporting evidence is not convincing. It is well known that fine-tuning and instruction-prompting can significantly impact a model's performance.**
>
> First, we wish to draw your attention to the fact that $g$ itself is not the result of fine-tuning or instruction-prompting from $a$. It is the model obtained as $g(y) = \frac{a(y)b(y)}{Z}$, with $Z$ a normalization constant. Given this formulation, we note that on the support of $g$, corresponding to the set of sequences satisfying constraint $b$ (i.e., the $y$'s such that $b(y) = 1$),  we have $g(y) = a(y) / Z$. Intuitively, this means that $g$ is *proportional* to $a$ on the support of $g$.
>
> Now, let us consider again a math problem which excludes political parties A, B, and C, as discussed in our previous response. If we have a set of numerical answers for this math problem, these answers will be ranked in the same way by $a$ and $g$ due to the proportionality relationship presented above. In other words, if $a$ is able to identify the correct answer among the set of possible answers, then $g$ will also identify the same correct answer. This proves that the performance of $g$ and $a$ are *equivalent* on the support of $g$ for this example, and this can be easily extended to more general cases. We will clarify this property in the paper when updating is allowed again.
>
> **Q4. While I appreciate the insights, the need to generate such a large number of samples for a relatively simple constraint amplifies my concerns about the practicality and scalability of the approach.**
>
> In our previous answer, we mentioned that we experimented with up to 800K and 1.5M samples from $a$ for our two settings, which indeed may appear to be large numbers. However, we wish to remind that in practice the warm-start DPG variant is able to get a close-to-optimal $KL(g || a')$ for as little as 100K samples in the lexical constraint scenario and 200K samples on sentiment reversal (see Figs. 3 and 5, respectively). We believe that these numbers of samples remain in a reasonable range.
>
> Moreover, our experiments have been conducted in settings with particularly low acceptance rates (0.0023 for the lexical constraint scenario and 0.005 for sentiment reversal) to better understand the characteristics of guaranteed generation. There is however a large range of concrete applications where constraints such as safety, politeness, or cultural sensitivity would typically lead to a higher acceptance rate. GUARD would show great practical potential for such cases and would not suffer from scalability issues.
>
> Considering the additional elements we provided in response to your latest concerns, do you still believe that a score of 5 is the most appropriate for our work?

---

> > ### Comment · Reviewer_hVqd · 2024-12-02
> >
> > Thank you for the response. Here are my remaining concerns, which lead me to conclude "*that a score of 5 is the most appropriate*" for your work:
> >
> > **Q1:** Your explanation reinforces my initial concern that there is no guarantee that each method (specifically one of your proposed methods CAP) minimizes $KL(g || a')$, even though you stress its importance for both quality and efficiency. Also, not having a systematic way of finding a prompt that *tends to satisfy the constraint* places significant responsibility on practitioners.
> >
> > **Q3:** Let me rephrase my concern: the model you get through fine-tuning and instruction-prompting might perform worse than the original model (see my initial question). In your argumentation, it seems that you do not account for the instruction prompt $p$. Assuming that $x$ represents the user prompt, the fine-tuned model that is additionally conditioned on the instruction prompt $g'(y | x, p)$ is not proportional to the original model $a(y | x)$. As you show, even for relatively simple constraint problems, $p$ can resemble something like: "*Write sentences with the given words. diagnosis: Assessment of microscopical and clinical parameters in the diagnosis of diabetes mellitus. pandas: Column headings differ in spreadsheet that is merged with pandas data. change: How to change the decimal separator in MS Word? amazing:*", which can misguide the model.

---

> > > ### Author Response · Authors · 2024-12-04
> > > **Response to Reviewer hVqd**
> > >
> > > Thank you for your additional comments. We address them in the clarifications below.
> > >
> > > **Q1:**
> > >
> > > As formally outlined in our initial response to your concern regarding the minimization of $KL(g || a’)$, SFT and DPG are training-based techniques which directly aim at minimizing $KL(g||a')$. In contrast, CAP is a training-free baseline that uses the LLM's generalization ability but does not minimize $KL(g||a')$. On the other hand, CAP can reliably increase the acceptance rate (AR) without training. This insight led us to propose warm-start DPG as our main algorithm, where we use $a(\cdot|CAP)$ in the very early training stage of DPG to avoid inefficient early training due to a low AR, then switch to directly optimizing $KL(g||a')$. As is already the case for vanilla DPG, the warm-start DPG algorithm is theoretically guaranteed to reduce $KL(g||a')$ and thus improve $a’$.
> > >
> > > While studying novel prompt-tuning methods to minimize $KL(g||a')$ could be meaningful, we did not include it within the scope of this paper since we focus on warm-start DPG as a novel training algorithm.
> > >
> > > **Q3:**
> > >
> > > We agree with you that using CAP on its own compromises the original model's capabilities because it increases $KL(g||a')$, as demonstrated in our experiments and analysis. For this reason, what we advocate is to use it *only* for warm-starting DPG, to benefit from its larger acceptance rate. You are also right that our goal should be to obtain a $g’$ that minimally distorts $g$, in other words that $KL(g||g’)$ should be small. Indeed, as mentioned in our previous response, $g$ is the model that does not distort at all the capabilities of $a$ on outputs that satisfy the constraint, so it is really the target we are aiming for.
> > > It is also true that the choice of prompt may influence this distortion. So it would be a valuable follow-up topic to explore the landscape of prompts with this objective in mind. Concerning the specific long prompt that you mention, it was included in Table 4 as an example of a few-shot prompt, providing examples for including keywords such as ‘diagnosis’, ‘pandas’, or ‘change’ in the next sentence, and asking the model to produce a sentence containing ‘amazing’. This prompt gave a $KL(g||a’)$ of 22.98 (which is indeed quite bad) while the best pale-blue dot in Fig. 4 had a KL of 9.88, and corresponded to the very simple and “natural” first prompt in Table 4, namely “*Next sentence should contain ‘amazing’*”. Pending more thorough exploration of different prompt strategies, it would seem like a good heuristics to aim for prompts which express in the most direct way possible the intention of the constraint.
> > >
> > > We sincerely hope this clarifies your concerns.

---

### Official Review · Reviewer_2U8J · 2024-11-04

**Soundness:** 3
**Presentation:** 3
**Contribution:** 3
**Rating:** 8
**Confidence:** 3

**Summary:**

This paper focuses on how to guarantee generation of restricted text that satisfy some given constraints from large language models, while preserving distribution information in original model as much. Specially, this paper first proposed a theorem showing there exist distributions that are impossible to be fitted by a regular autoregressive model, so that sampling strategy should always be combined with. Then an algorithm named GUARD is proposed for this task. It first finetunes the model for higher acceptance rate, and then applies rejection sampling for strict guarantee. Experiments on two constrained generation tasks show that, GUARD achieves significantly higher acceptance rate, while keeping close to the ground truth distribution.

**Strengths:**

Guaranteed generation is an important requirement in real applications of LLMs. Though relatively easy to come up with, the algorithm proposed in this paper do provide an effective solution for this problem. Meanwhile, Theorem 1 gives a good reminder for trials attempting to solve such problem without considering sampling strategies. Overall, I would be glad to see this paper being accepted, as an important progress under this specific problem.

**Weaknesses:**

While this paper has its merit in terms of contribution, it did not give me a good impression at first glance. The following suggestions may be helpful to the authors:
1. Algorithm 1 is too short. It is frustrating to see a 3-line algorithm in a paper. Switching its position with that of Algorithm 2 would be much better.
2. The content of Theorem 2 is too simple to be a theorem. It's OK to put it as an equation.
3. For Theorem 1 in main text, I suggest to replace it with the complete version in appendix. The concept of PTC is central and should be highlighted in the main text.
4. In section 2.3, It may not be a good idea to discuss the non-zero probability under softmax as an evidence for limits of autoregressive models, since it is not the key point and such shortcoming can be easily avoid by rejection sampling. I suggest to discuss the PTC property of common models instead, which will serve for Theorem 1.
5. In experiments, I did not find the exact numbers of AR of DPG. I notice that they are reported in the figures by coordinates, but exact numbers are also necessary.
6. Text in figures and tables are too small to be viewed on A4 paper, please consider rearranging the layout.

**Questions:**

1. How do you get the gold samples in Fig. 4&7?
2. I notice that you discussed the relation to I-projection in line 136&242. How do such discussions help? I found that removing these contents does not affect understanding.

---

> ### Author Response · Authors · 2024-11-20
> **Response to Reviewer 2U8J**
>
> Many thanks for your very positive feedback and for the useful suggestions for improvement! We address below the points raised in your review.
>
> **Suggestion 1. Algorithm 1 is too short. It is frustrating to see a 3-line algorithm in a paper. Switching its position with that of Algorithm 2 would be much better.**
>
> Thank you for the suggestion. We agree that the way Algorithm 1 is presented may give a bad first impression to the reader, and we will update the submission to present it in a more subdued way inside the text. As for Algorithm 2, it would take a lot of space in the main text, is focused on training, not on sampling, and its exact form is actually less essential to the core message of the paper than what the extremely simple Algorithm 1 says, so we prefer to keep it in the Appendix.
>
> **Suggestion 2. The content of Theorem 2 is too simple to be a theorem. It's OK to put it as an equation.**
>
> It is true that the formulation of this theorem is simple, as well as its proof (see Appendix A.6 for two proofs, including one simple direct derivation). However, its proof is not the most important part, but rather its interpretation, which is core to our work. While we have considered calling it a “Lemma” or a “Fact”, we prefer not to give it such a secondary status in relation with Theorem 1.
>
> **Suggestion 3. For Theorem 1 in main text, I suggest to replace it with the complete version in appendix. The concept of PTC is central and should be highlighted in the main text.**
>
> We totally agree with this suggestion and will follow it, thanks a lot!
>
> **Suggestion 4. In section 2.3, It may not be a good idea to discuss the non-zero probability under softmax as an evidence for limits of autoregressive models, since it is not the key point and such shortcoming can be easily avoid by rejection sampling. I suggest to discuss the PTC property of common models instead, which will serve for Theorem 1.**
>
> We will discuss the PTC property, as mentioned above in relation to Theorem 1, but also find the discussion of more mundane properties of autoregressive models useful, and they also connect, here and in the Appendix that expands on them, with some remarks by other reviewers.
>
> **Suggestion 5. In experiments, I did not find the exact numbers of AR of DPG. I notice that they are reported in the figures by coordinates, but exact numbers are also necessary.**
>
> Thank you for pointing out what we missed. We will explicitly provide the AR values before and after training.
>
> **Suggestion 6. Text in figures and tables are too small to be viewed on A4 paper, please consider rearranging the layout.**
>
> We are sorry that the paper may be difficult to view on A4 paper, and we do agree that formatting needs improvement, but this is unfortunately difficult to do while remaining within the 10 pages limit. We will do our best to maximize the use of the free space that remains to address feedback from reviewers and to improve the layout.
>
> **Question 1. How do you get the gold samples in Fig. 4&7?**
>
> By applying Algorithm 1 with $a’=a$ (see second paragraph of Section 3) it is possible to obtain samples from $g$ by first sampling from $a$ and then filtering with $b$ (though this may be highly inefficient in case of low $AR_a$).
>
> **Question 2. I notice that you discussed the relation to I-projection in line 136&242. How do such discussions help? I found that removing these contents does not affect understanding.**
>
> We agree with you that, concerning line 242, the reference to I-projection is not absolutely necessary, because it is possible to give a simple self-contained derivation (we do so in Appendix A.6) and we will reformulate the text to explicitly mention that derivation. However, we feel that mentioning I-projection in the context of Theorem 2 could be useful for extensions of the work presented here (I-projection is an important information-theoretic concept, that can also be used for soft constraints of the type used in Khalifa et al (2021)).
>
> On the other hand, in the context of line 136, we feel that this notion is really useful, because it provides another motivation for the definition of $g$, namely as the distribution that minimally distorts $a$ while respecting the constraint. So, overall, despite the introduction of not absolutely necessary terminology, we prefer to keep it.

---

> > ### Author Response · Authors · 2024-11-24
> > **Follow up to our response to Reviewer 2U8J**
> >
> > Dear Reviewer 2U8J,
> >
> > As the end of the discussion period is nearing, we would like to follow up to see if our detailed response addressed your concerns. We would also be happy to answer any further questions you have. Thank you again for your time and helpful feedback.

---

> > > ### Comment · Reviewer_2U8J · 2024-11-28
> > >
> > > Thank you for your response. I have read the rebuttals and comments from other reviewers. I decide to keep my rating.

---

> > > > ### Author Response · Authors · 2024-11-28
> > > > **Response to Reviewer 2U8J**
> > > >
> > > > Thank you for taking the time to read our response as well as our discussions with other reviewers, and thank you again for your very positive review.

---

### Official Review · Reviewer_DNMP · 2024-11-10

**Soundness:** 2
**Presentation:** 3
**Contribution:** 1
**Rating:** 3
**Confidence:** 5

**Summary:**

This paper motivates and studies the problem of constraining LLM generation on logical constraints with guarantees. The authors show that it is often intractable to directly sample from the LLM distribution conditioning on a logical constraint and propose the GUARD framework for solving this problem. GUARD performs rejection sampling on an approximate autoregressive distribution of the desired conditional distribution, which can be obtained via supervised fine-tuning, prompting or distributional policy gradient (DPG). The authors evaluate GUARD on two tasks: (1) generate a piece of text of 30 tokens while including the string "amazing" and (2) generate a story ending of positive sentiment given a story beginning of negative sentiment. The authors evaluate the aforementioned three alternatives for obtaining the approximate distribution.

**Strengths:**

The papers motivate an important challenge for the existing LLMs and provide proofs that it is theoretically intractable to sample from the LLM distributions conditioned on logical constraints.

**Weaknesses:**

The authors could have done a more extensive literature survey and demonstrate how their work relates to/differs from following approaches for constrained generation:

FUDGE [1]: trains auxiliary classifiers on existing classification datasets and uses the classifiers to guide generation to satisfy the constraint.

NeuroLogic A*esque Decoding [2]: performs lookahead decoding with heuristic functions that estimate how likely the constraint will be satisfied.

NADO [3]: trains auxiliary classifiers on data sampled from LLMs and uses the classifiers to guide generation to satisfy the constraint.

GeLaTo [4]: uses an Hidden Markov Model to enforce the constraint

Outlines [5]: uses regex/finite-state machines to mask out next tokens that would violate the constraint.

In particular, the authors claim that they are not aware of general techniques that (1) guarantees the constraint is enforced and (2) limiting the distortion of the original distribution while such techniques, or related techniques that partially achieve (1) or (2), exist in literature: [4,5] achieves (1) and [1,2,3,4] tries to optimize for (2).

Similarly, for both tasks considered in the experiment section, the authors could have instead use existing benchmarks: for the task of generating text using given keywords, the authors could consider the CommonGen dataset [6] and compare their approach against [2,3,4].
For the task of sentiment control, the goal is to generate text such that an existing classifier assigns a score > some threshold. In this way, the authors might as well evaluate their approach on the task of formality control by replacing the sentiment classifier with a formality classifier and compare against [3]. Or the authors could consider the task of topic control and compare against [1].

[1] Kevin Yang and Dan Klein. 2021. FUDGE: controlled text generation with future discriminators. In NAACL-HLT. Association for Computational Linguistics.

[2] Lu, X., Welleck, S., West, P., Jiang, L., Kasai, J., Khashabi, D., ... & Choi, Y. (2021). Neurologic a* esque decoding: Constrained text generation with lookahead heuristics. arXiv preprint arXiv:2112.08726.

[3] Meng, T., Lu, S., Peng, N., & Chang, K. W. (2022). Controllable text generation with neurally-decomposed oracle. Advances in Neural Information Processing Systems, 35, 28125-28139.

[4] Zhang, H., Dang, M., Peng, N., & Van den Broeck, G. (2023, July). Tractable control for autoregressive language generation. In
International Conference on Machine Learning (pp. 40932-40945). PMLR.

[5] Willard, B. T. & Louf, R. Efficient Guided Generation for Large Language Models. arXiv preprint. arXiv: 2307.09702 [cs.CL] (2023).

[6] Lin, B. Y., Zhou, W., Shen, M., Zhou, P., Bhagavatula, C., Choi, Y., & Ren, X. (2019). CommonGen: A constrained text generation challenge for generative commonsense reasoning. arXiv preprint arXiv:1911.03705.

**Questions:**

I wonder if the distribution used by [3] would be a better proposal distribution for rejection sampling, compared to SFT, CAP or DPG.

---

> ### Author Response · Authors · 2024-11-20
> **Response to Reviewer DNMP (part 1)**
>
> Thank you for your feedback, and in particular for calling our attention to some important references that we had missed at the time of submission.
>
> We feel that [4] is the one most directly related to our work, and we will start by discussing it, stressing common points as well as fundamental differences, before moving to other aspects of your feedback. We will also update our submission (as soon as possible before the rebuttal deadline) with expanded related work reflecting [4] and other references, and will also correct mistaken statements implying that our work is the first to address 100% constraint satisfaction while attempting to keep close to the original LLM.
>
> **Relation of our work to [4]**
>
> Similar to us, [4] is able to produce outputs that always satisfy the constraint, while maintaining proximity to the original model. It does so by approximating the LLM with an HMM, which, in contrast to the LLM, supports a dynamic programming approach to complying with lexical constraints over the output, and therefore the ability to weigh the long-term consequences of local next-token choices relative to these constraints. When the HMM approximation to the LLM is good enough (which depends in particular on the number of states of the HMM), the same HMM can be used at inference time with different lexical constraints without need of retraining.
>
> We acknowledge the interest of this approach, but note several fundamental differences with what we do.
>
> - We handle *arbitrary logical constraints* (that is, binary predicates), not necessarily of a lexical nature, which would be difficult to handle through HMMs or similar finite-state mechanisms.
> - We give a central status to the “gold” constrained distribution $g$, which is the distribution minimally deviating from the original distribution while still fully satisfying the constraint, and we evaluate the quality of our sampler in terms of distributional distance to this distribution $g$. [4] does not focus on the evaluation of a *sampler* relative to a reference distribution, but rather on the evaluation of a *decoder* in terms of downstream tasks which have a looser relation with the constraint.
> - We study the trade-off between the quality of the sampler and its efficiency in the context of a simple rejection sampling mechanism based on an autoregressive approximation $a’$ of $g$ and show that quality and efficiency are both directly controlled by the divergence $KL(g||a’)$. This in turns motivates our interest for an approximation technique that focuses on minimizing this divergence, such as DPG.
>
> Note: Apart from [4], we recently came across the follow-up work [7], which generalizes [4] by considering lexical constraints defined through DFA’s (Deterministic Finite Automata). As a side note, on p. 13 of our original appendix (“Solvable instances of the problem”), we mentioned briefly that in case the base autoregressive model and the lexical constraint are based on weighted finite state automata, the intersection of these automata could be constructed and sampled from. Such weighted automata are very much related to HMMs, of course, and we will also update the appendix to refer to [4,7], which had escaped our attention at that time.
>
> [7] Zhang, H., Kung, P., Yoshida, M., Van den Broeck, G., & Peng, N. (2024. August). Adaptable Logical Control for Large Language Models. arXiv preprint arXiv:2406.13892.

---

> ### Author Response · Authors · 2024-11-20
> **Response to Reviewer DNMP (part 2)**
>
> **Relation to other works mentioned**
>
> Thank you for the other references you mentioned. There is indeed a very large literature on controlled generation and we cited a survey from 2023. Apart from [4], the only such reference concerned with strict guarantees is [5], also in the context of lexical constraints, but here the focus is simply on efficient pruning of next tokens that do not satisfy a finite-state constraint, without any global reweighing. Although we were not aware of this work at the time of submission, we did discuss a similar approach for enforcing or avoiding the inclusion of specific keywords in Appendix A.2. We will update this section to add a reference to [5].
>
> Concerning the other references, note that we did actually cite [1], which is indeed relevant to our work, and we will also cite [3], which does have some significant relation to our work. We will mention [2] as well, although it is strongly focused on deterministic decoding rather than sampling.
>
> Concerning the CommonGen challenge [6], the constraints correspond to lists of keywords, and the task is to formulate sentences that contain the keywords and follow a certain common-sense logic. In particular, the dataset includes a few specific target sentences for each list of keywords, which are provided as references. The goal for evaluated models is then to generate sentences as close as possible to these few references. This substantially differs from our distributional objective where we seek to generate sentences that are distributed in a similar way as samples from $g$. For that reason, it does not seem straightforward to evaluate GUARD on CommonGen.
>
> **Using NADO distribution [3] as proposal in GUARD**
>
> The NADO distribution used by [3] is trained with an objective similar to ours, namely it tries to approximate a distribution comparable to our gold filtered distribution $g$. As a training method to obtain an autoregressive distribution similar to our $a’$, it has some analogies with (i) SFT, by training on samples from the filtered distribution, and (ii) DPG, by also performing a kind of distributional matching, but with more emphasis on local (i.e., token-level) decisions. Contrary to DPG, NADO does not directly have the objective of minimizing the divergence $KL(g||a’)$, which is the determining factor for the success of GUARD. Using NADO for $a’$ would nonetheless be interesting as a follow-up work, to see whether the value of its divergence is competitive with the cases we have explored, leading to improved performance for GUARD.

---

> > ### Comment · Reviewer_DNMP · 2024-11-24
> >
> > Thank you for clarifying how your work relates to the other ones.
> >
> > Rejection sampling does handle arbitrary constraint, given unlimited computation resources and unlimited time. However you did provide much insight to the real question: for which constraints is rejection sampling able to give a good (no matter how you define it) approximation to the original distribution in a reasonable amount of time? A systematic empirical/theoretical analysis about this question would be important.
> >
> > You mentioned that your only goal is to match the original distribution as closely as possible, so it does not make sense to evaluate via the more indirect metrics such as the BLEU score. However, it is practically infeasible to measure the KL divergence as the ground truth (the LLM distribution conditioning on some constraints) is computationally intractable to sample from or estimate, even for extremely simple constraints. How are you able to demonstrate the effectiveness of your approach when the metric itself can only be indirectly estimated?

---

> > > ### Author Response · Authors · 2024-11-24
> > > **Response to Reviewer DNMP**
> > >
> > > Thank you for taking the time to get back to us and for acknowledging that our previous response addresses your concerns about the positioning of our work with respect to existing literature. We address your additional questions below.
> > >
> > > **Q1: For which constraints is rejection sampling able to provide a good approximation in a reasonable time?**
> > >
> > > The discussion point you mentioned (approximation to the original distribution in a reasonable amount of time, i.e., with a high acceptance rate) is indeed one of the main focuses of this paper. As stated in the last line of the first paragraph of the Introduction, one of our key research questions is: "How can we simultaneously obtain the two previous properties *(guaranteed constraint satisfaction and distributional closeness)* at a limited inference cost *(expressed by the acceptance rate)*?”. This is formalized in Theorem 2, which shows that our training objective $KL(g || a')$ can be decomposed as the sum of the distributional closeness to the target $g$ (i.e., $KL(g || g')$) and the negative log of the acceptance rate of $a’$ (i.e., $- log AR_{a'}$).
> > >
> > > From an empirical standpoint, in the two settings considered in the paper, we show that using naive rejection sampling on the *original* distribution $a$ leads to a very poor efficiency: for the lexical constraints scenario and the sentiment reversal scenario, the acceptance rates $AR_a$ are respectively 0.0023 and 0.005. In more intuitive terms, using $a$ for rejection sampling leads to accepting 1 sample out of 435 drawn samples for lexical constraints and 1 sample out of 200 drawn samples for sentiment reversal. In comparison, using GUARD, this goes up to accepting approximately 1 sample out of 2 drawn samples and 1 sample out of 3 drawn samples, respectively (i.e., 180x and 50x improvement in the acceptance rate with respect to the naive rejection sampling approach, as detailed in Sections 4.1 and 4.2). These numbers show that using GUARD enables a reasonable efficiency for the two use-cases considered.
> > >
> > > Naturally, in a scenario where the acceptance rate of $a$ is much higher (e.g., 0.5), using rejection sampling directly on $a$ is enough as it would already lead to satisfactory efficiency. We indicated this in the second paragraph of Section 3, where we pointed out that GUARD is most beneficial for *difficult* constraints, i.e., constraints which are rarely satisfied by $a$.
> > >
> > > We also agree that it would be interesting to analyze the impact of the nature of a constraint on the ability to approximate $g$ within a large-scale study that considers many different constraints. However, we believe that such an analysis is beyond the scope of this paper, which is focused on the introduction of GUARD and its theoretical properties for any constraint $b$.
> > >
> > > **Q2: As $g$ is computationally intractable to sample from or estimate, how are you able to demonstrate the effectiveness of your approach when the metric itself can only be indirectly estimated?**
> > >
> > > According to Theorem 1, it is indeed typically impossible to find an autoregressive model that matches distribution $g$ exactly, but we can still obtain gold samples from $g$ by first sampling from $a$ and then filtering with $b$ (though this may be highly inefficient in case of low $AR_a$, as discussed above). These samples are used to estimate $KL(g || g')$ as detailed in Equation 10 in Appendix C. The estimate of $KL(g || a')$ is obtained in the same way. Similarly, $AR_{a'}$ which is equal to $\mathbb{E}\_{y \sim a'} [b(y)]$ can be estimated by drawing samples from $a’$. Given that $Z = AR_a$ and $Z’=AR_{a’}$ (see Appendix A.5 for more details), the same technique can be used to estimate $Z$ and $Z’$ in Equation 10. We will update the paper to clarify the estimation of these different quantities.
> > >
> > > In practice, we draw 1,000,000 samples from $a$ for the KL estimation. In the lexical constraints scenario (where $AR_a = 0.0023$), this leads to approximately 2,300 samples from $g$. For the sentiment reversal experiment (where $AR_a = 0.005$), this yields 5,000 samples from $g$. Such large numbers of samples from $g$ provide a good basis for the estimation of the KL metrics, which is also underscored by the consistency of the results across our two settings as showcased by Figures 3 and 6.
> > >
> > >
> > > Thanks again for your time, and please let us know if you have any further questions.

---

> > > > ### Author Response · Authors · 2024-11-29
> > > > **Gentle reminder before the end of the discussion period**
> > > >
> > > > Dear Reviewer DNMP,
> > > >
> > > > Thank you once again for taking the time to react to our initial responses. As the end of the discussion period is approaching and it has already been a few days since we have posted our follow-up responses, we would like to know if we have addressed the concerns you raised in your latest questions. If you feel that it is the case, we would be very grateful for this to be reflected in score adjustments to acknowledge our responses. In particular, we would like to check with you if your initial score of 1 for the contribution criterion still reflects your current assessment of our submission, in light of our different responses.

---

> > > > > ### Comment · Reviewer_DNMP · 2024-11-30
> > > > >
> > > > > Thank you for your response. My initial score of 1 still reflects my current assessment of your submission.
> > > > >
> > > > > Q1. If you believe that studying the effectiveness of rejection sampling in approximating different families of constraints is out of the scope of your paper then I believe that the scope of your work is too limited for it to be accepted. (1) the concept of achieving controllable generation while minimizing distribution shifts (i.e. approximating the desired conditional distribution) has long existed in literature and (2) the proposal to use vanilla rejection sampling without in-depth analysis (either empirical or theoretical) of its behavior should not be considered enough technical contribution.
> > > > >
> > > > > Q2. Again, when you mentioned in practice you were able to draw 1M examples to estimate the KL divergence, that is only because the lexical constraint being tested on is a trivial one. If I'm wrong about this point, please correct me by showing me the procedure for estimating the KL-divergence (or equivalently the ground-truth normalization constant) for the constraint that "amazing", "demand", "tragedy" all appearing in the first 50 generated tokens of the LLM.

---

> > > > > > ### Author Response · Authors · 2024-12-04
> > > > > > **Response to Reviewer DNMP (part 1)**
> > > > > >
> > > > > > Thank you for your response, we are happy to clarify the new issues you raise.
> > > > > >
> > > > > > **Q1:**
> > > > > >
> > > > > > We think there is a misunderstanding here. We consider that what is out of scope in our work would be a large-scale study on a great number (e.g., hundreds or thousands) of constraints, and the impact of the different types or families of constraints on the acceptance rate — and thus, on the performance of applying rejection sampling from $a$. Studying the effectiveness of GUARD in approximating different families of constraints *is* in the scope of our paper, and our experiments include a positive ending constraint and lexical constraints (with the keyword “amazing” in Section 4.1, as well as additional analysis for keywords “restaurant” and “world” in Appendix E, Fig. 9).
> > > > > >
> > > > > > (1) While we agree that there exists significant literature on controlled text generation for approximating a given distribution, a primary differentiator in our work is the *simultaneous* enforcement of strict constraint satisfaction. To the best of our knowledge, the only existing works that simultaneously address both dimensions are [4] and [7] which we have already discussed extensively in our initial response and the revised version of our submission. However, as previously explained, [4, 7], which have a different perspective than us in terms of formal objective, are limited to lexical constraints and cannot, for instance, enforce the positive ending constraint described in Section 4.2. This reveals a clear gap in the literature: the lack of a general approach for enforcing strict constraint satisfaction beyond lexical constraints, while remaining close to the original distribution. Our paper seeks to address this gap, with both theoretical and empirical contributions as highlighted in our last message below (and summarized in our general message to all reviewers).
> > > > > >
> > > > > > (2) One key point in the paper is in fact that if we are able to approximate $g$ with an autoregressive model $a'$, then the simple form of rejection sampling that we describe inherits great properties from $a'$. To reiterate, that is the content of Theorem 2, namely $KL(g||a’) = KL(g||g’) - \log AR_{a'}$. In words, it says that if $KL(g||a’)$ is small, then on the one hand the rejection sampler will be efficient (i.e., $AR_{a'}$ will be close to the optimal acceptance rate of 1), and on the other hand, the obtained sampler $g’$ will be close to $g$ (i.e., $KL(g||g’)$ will be small). We believe this theorem (and its consequences) to be novel and an important theoretical contribution of our work that would deserve acknowledgment. Concerning the empirical study of the proposed approach, we do experimentally contrast several (admittedly, non-exhaustive) approximation techniques (SFT, DPG, CAP). We compared not only their ability to obtain a good $KL(g||a’)$, but also the trade-offs they introduce between $KL(g||g’)$ and $AR_{a'}$. For instance, we discussed the fact that CAP methods are able to produce reasonable acceptance rate but at the cost of a large $KL(g||g’)$ (see Fig. 4 and 7), leading to distortion in both sentiment level distribution and keyword position from our analysis (see Fig. 2 and 6). We once again believe that such empirical analysis presents some meaningful value.

---

> > > > > > > ### Author Response · Authors · 2024-12-04
> > > > > > > **Response to Reviewer DNMP (part 2)**
> > > > > > >
> > > > > > > **Q2:**
> > > > > > >
> > > > > > > In a previous question, you expressed doubts about the feasibility of estimating this divergence “even for extremely simple constraints”. We answered that it was actually possible for constraints which do not have an overly low initial acceptance rate $AR_a$, pointing to a simple technique detailed in the paper (and further elaborated in its revised version). In that regard, we wish to point out that while the constraints included in the paper lead to a non-negligible $AR_a$ (e.g., 0.0023 and 0.005 for the two main experiments), many practically relevant constraints — such as those related to safety, politeness or sentiment polarity — will also exhibit a sufficiently high $AR_a$ for our estimation technique to be applicable. We also note that such constraints, which are typically non-lexical, encompass a concept of *non-trivial* complexity while having a reasonably high $AR_a$. Thus, we wish to emphasize that high constraint complexity and practical value do not necessarily imply a very low acceptance rate.
> > > > > > >
> > > > > > > Regarding your current question, you are asking about a non-trivial lexical constraint — whose complexity is in fact only tied to its much lower acceptance rate. In that case, if we wanted to compute any of the quantities $Z$, $KL(g||a’)$, $KL(g||g’)$, the approach that would consist in first producing a large sample from $a$, and filtering by $b$ to obtain a sample from $g$ would indeed be inefficient due to the small $AR_a$ (as we have already acknowledged in our previous response). We should note that, for such a constraint, and for *any autoregressive model* $a’$ (including $a$), *independently of the approach taken in GUARD*, the problem of estimating $KL(g||a’)$ is a difficult one.
> > > > > > >
> > > > > > > We thus believe that proposing an exhaustive solution to this problem falls outside the scope of our submission. However, in an attempt to provide some elements of a response to you, we sketch one possible — preliminary — approach. This approach is related to the warm-start version of the DPG algorithm (Algorithm 2 in the Appendix), which we have already advocated for the relatively rare constraints that we experimented with. The proposed approach can be outlined as follows:
> > > > > > >
> > > > > > > 1. Initialize warm-start DPG with a CAP prompt such as “Here is a text containing the tokens ‘amazing’, ‘demand’, ‘tragedy’ in its first 50 tokens, in any order:“.
> > > > > > > 2. At the end of the CAP initialization phase of Algorithm 2, we obtain a first model $\pi_\theta$ (line 10), which should satisfy the constraint much more often than $a$ (this is something that we have observed in our experimental settings and is intuitive, but it would need to be confirmed in this case).
> > > > > > > 3. We then start the DPG fine-tuning on line 12, with the proposal $\pi_\theta$. The fact that the $y$’s produced on line 17 often respect the constraint leads to effective gradient steps on line 21 (because $p(y)$ is frequently non-zero).
> > > > > > > 4. At the end of the fine-tuning process, we obtain an $a’=\pi_\theta$ which has a better $KL(g||a’)$ than the initial $\pi_\theta$ — as this is the loss that DPG is trying to minimize (see Appendix B.7 added in the revised PDF). Also, as a by-product of the training process, we obtain an estimate of $Z$ on line 19 (it can be easily proven that the value produced on line 19 is an unbiased estimate of the true $Z$, and we will add the proof to the revised version of the paper when we can edit it again).
> > > > > > > 5. At this point, we can expect to have a model $a’$ with a reasonably low $KL(g||a’)$ and thus with a relatively high $AR_{a’}$ (given the relationship between the two quantities established in Theorem 2). We also have a reasonable, unbiased estimate of $Z$. One way to estimate $KL(g||a’)$ is then by noting that $KL(g||a’) = \mathbb{E}\_{y \sim g} \log\frac{g(y)}{a’(y)} = \mathbb{E}\_{y \sim a’} \frac{g(y)}{a’(y)} \log\frac{g(y)}{a’(y)}$. The last equality is obtained by using importance sampling to replace the expectation relative to $g$ by an expectation relative to $a’$ which is much easier to sample from, given the higher acceptance rate $AR_{a’}$ compared to $AR_{a}$. The quantity $g(y) = \frac{a(y)b(y)}{Z}$ can also be easily approximated based on the previously obtained estimate of $Z$. The other quantities $AR_{a’}$ and $KL(g||g’)$ can be computed in similar ways.
> > > > > > >
> > > > > > > All of this, obviously, would have to be confirmed experimentally. However, we hope the approach outlined here offers some insights into your request regarding the procedure for estimating the KL-divergence in the case of multi-keyword constraints (and, more generally, any constraint with an extremely low acceptance rate that can be somehow expressed through a prompt).

---

> > > > > > > > ### Author Response · Authors · 2024-12-04
> > > > > > > > **Final remarks to reviewer DNMP regarding our paper**
> > > > > > > >
> > > > > > > > We hope that the responses in our previous messages were helpful to address your concerns. More generally, we would like to respectfully suggest that you reconsider whether the contributions listed in our general message to all reviewers (that we have just posted) deserve a score of 1. In the *Strengths* rubric, you mentioned only one positive point related to Theorem 1, without addressing other aspects of our work. For instance, you did not comment — positively or negatively — on our Theorem 2, nor on our experimental results. We firmly believe that Theorem 1, Theorem 2, and our experimental findings, on their own, would merit a higher contribution score.
> > > > > > > >
> > > > > > > > While we may not be in agreement with the assessed value for our work, we nonetheless wish to thank you for your high engagement in this discussion and for mentioning to us, early in the interaction, relevant prior works that we had missed. We fully agree on the importance to cite and correctly differentiate from these works, as we hope we did. We also make this point clear in our general message.

---

### Author Response · Authors · 2024-11-20
**General message to all reviewers**

We thank the reviewers for their thorough reviews, helpful feedback, and suggestions for improvement. We are providing initial responses to each of you and will upload the final version of our submission before the end of the discussion period. We do this in order to facilitate interactions with you, which we are looking forward to, and also to make the final PDF update as relevant and useful as possible.

---

> ### Author Response · Authors · 2024-11-26
> **Updated PDF with revisions**
>
> Dear Reviewers,
>
> Thank you again for the time invested in our paper and for your valuable feedback!
>
> We have uploaded an updated version of our submission, in which we have attempted to address your detailed questions and suggestions. The changes are highlighted in blue for your convenience. We hope that these revisions will help clarify any concern you may have had about our paper and that they do improve the overall quality of our work.
>
> If you feel that our responses and revisions addressed your concerns, we would be most grateful for this to be reflected in score adjustments; otherwise, we would be happy to further discuss any remaining doubts.

---

> > ### Author Response · Authors · 2024-12-04
> > **Recap of our contributions and summary of the discussion**
> >
> > We would like to thank all reviewers again for their time, thorough work, and interactions with us relative to this paper.
> >
> > At this point, we would like to recap on how we now perceive our main contributions and what we have learnt from these interactions.
> >
> > One of our main contributions is at the level of the **conceptual perspective** that we take on the problem of guaranteed generation under a constraint $b$, namely by seeing it as the problem of designing a generator $g’$ that at the same time: (a) strictly satisfies the constraints, (b) minimizes the divergence $KL(g||g’)$ to the ideal distribution $g$ defined by the constraint, and (c) is efficient.
> >
> > We address this goal through the GUARD approach, that combines a training-time aspect with the purpose of obtaining an autoregressive model $a’$ such that $KL(g||a’)$ is minimized with an inference-time approach that performs a simple form of rejection sampling. By design, GUARD then guarantees that the constraint $b$ will be satisfied.
> >
> > The main **theoretical innovation** of our work is then to observe and prove a key property of our approach, namely Theorem 2, which relates in a very simple and interesting way $KL(g||a’)$ with $KL(g||g’)$ and the generator efficiency $AR_{a’}$ (i.e., its acceptance rate). To the best of our knowledge this property has never been stated before. A consequence is that the quality of the autoregressive approximation $KL(g||a’)$ directly controls the quality of the generator’s approximation $KL(g||g’)$ as well as its efficiency. It also implies that for the same level of $KL(g||a’)$, there is a direct trade-off between $KL(g||g’)$  and $AR_{a’}$.
> >
> > A second **theoretical contribution** is the statement and proof of the fact (Theorem 1) that, in general, no autoregressive model $a’$ can perfectly reach $g$, in other words, be such that $KL(g||a’)=0$. While this is, in its principle, a consequence of earlier work by Lin et al. (2020b), we provide a self-contained proof in the context of constraint satisfaction, and hope to make this important fact better known to the controlled generation community.
> >
> > On the **algorithmic side**, our main contribution is the introduction of the “warm-starting” extension for the DPG algorithm for training an $a’$ minimizing $KL(g||a’)$, which is a way to exploit prompts for accelerating the early training of DPG.
> >
> > On the **experimental side**, our main contributions consist of demonstrating the effectiveness of GUARD across a lexical constraint scenario and a sentiment reversal scenario with a positive ending constraint. We empirically verify that the GUARD algorithm can almost preserve the gold distribution through warm-start DPG combined with rejection sampling, while significantly increasing the $AR_{a’}$ over $AR_a$ — by a factor of 200 and 60, respectively. As a secondary outcome for this study, which may be of interest beyond GUARD, we also observed noteworthy shortcomings when prompts are used directly for approximating $g$. These include a tendency to have a high divergence with $g$ and to lower the diversity of the outputs, while a fine-tuning approach such as DPG does not have these defects (see Fig. 2 and 6, Tables 1 and 2).
> >
> > An important **improvement** that we made to the paper, based on the interactions, was to extend the **related work** section to include, in particular, the papers [Zhang et al (2023)] and [Zhang et al (2024)] which share our goal of strictly enforcing the constraints. This gave us the opportunity to explain differentiators between these papers and our work, in particular the fact that these papers, being based on finite-state mechanisms, are focused on lexical constraints. In other words, they cannot handle the more general type of constraint that we consider, such as the positive ending constraint of Section 4.2.
> >
> > Thanks to the feedback received from all reviewers, we have also (1) improved the formulation of Theorem 1, (2) tried to further motivate our focus on the effective use of constraint $b$ rather than on its design, (3) clarified the estimation procedure for evaluating KL divergences, and (4) generally tried to make the paper more helpful based on the discussion.

---

### Meta-Review · Area_Chair_9619 · 2024-12-22

**Metareview:**

## Summary

This paper addresses the challenge of ensuring strict constraint satisfaction in text generation by large language models while maintaining closeness to the original model’s distribution. The authors define an ideal distribution that satisfies the constraints and prove it cannot be achieved using autoregressive training alone. To address this, they propose GUARD, a method combining autoregressive proposal distributions with rejection sampling. Theoretical analysis shows that GUARD balances constraint satisfaction, distributional fidelity, and inference efficiency by minimizing KL divergence. Experimental results on lexical constraint generation and sentiment reversal demonstrate that GUARD achieves perfect constraint satisfaction with improved efficiency compared to other methods.

## Decision

The paper aims to address an important challenge for the existing LLMs and provide theoretical results related to this problem. The approach proposed in this paper provides an effective solution to this problem. The results are overall convincing. Overall, some of the reviewers had concerns about the novelty and not covering some of the relevant papers that should be cited, but it is not. I strongly encourage the authors to cite those relevant papers. I recommend the authors address all the concerns of the reviewers, including the citations for the camera-ready version of the paper.

**Additional Comments On Reviewer Discussion:**

Overall, the main concern raised by the reviewers was related to the novelty of the method proposed in the paper and the lack of citations for some of the relevant work. Especially, Reviewer DNMP felt strongly about it. However, I agree that the authors should cite and discuss those papers, but I think this is indeed easily fixable. I think the proposed approach is different enough from some of those works that I think this paper might be still worthwhile for acceptance.

---

### Decision · Program_Chairs · 2025-01-22

Accept (Poster)